# Dislocation Majorana bound states in iron-based superconductors

Lun-Hui Hu ®[1,2,3] & Rui-Xing Zhang ®[1,2,4] ✉

We show that lattice dislocations of topological iron-based superconductors such as $FeTe_{1-x}Se_x$ will intrinsically trap non-Abelian Majorana quasiparticles, in the absence of any external magnetic field. Our theory is motivated by the recent experimental observations of normal-state weak topology and surface magnetism that coexist with superconductivity in $FeTe_{1-x}Se_x$, the combination of which naturally achieves an emergent second-order topological super-conductivity in a two-dimensional subsystem spanned by screw or edge dis-locations. This exemplifies a new embedded higher-order topological phase in class D, where Majorana zero modes appear around the "corners" of a low-dimensional embedded subsystem, instead of those of the full crystal. A nested domain wall theory is developed to understand the origin of these defect Majorana zero modes. When the surface magnetism is absent, we further find that $s_\pm$ pairing symmetry itself is capable of inducing a different type of class-DIII embedded higher-order topology with defect-bound Majorana Kramers pairs. We also provide detailed discussions on the real-world material candi-dates for our proposals, including $FeTe_{1-x}Se_x$, LiFeAs, $\beta$-PdBi$_2$, and hetero-structures of bismuth, etc. Our work establishes lattice defects as a new venue to achieve high-temperature topological quantum information processing.

Crystals of quantum materials are rarely perfect in the real world. While it appears natural to always suppress lattice disorders and pur-sue crystals of a higher purity, defectiveness in topological quantum materials often binds exotic massless quasiparticles that hold great promise for future electronics. A prototypical example is the famous Jackiw-Rebbi problem[1] and its condensed matter realization in polyacetylene[2], where zero-energy fermionic modes are trapped by the domain wall defects of a one-dimensional (1D) dimerized atomic chain. Since then, gapless electronic or Majorana zero modes have been established in lattice or order-parameter defects of various topological phases, including weak and crystalline topological insula-tors (TIs)[3-7], topological superconductors (TSCs)[8-15], and topological semimetals[16,17], etc. For example, locally irremovable lattice topologi-cal defects such as screw/edge dislocations can trap 1D helical bound states in 3D weak TIs, providing an intriguing bridge between lattice and electronic topologies. Experimental evidence for dislocation-

trapped electronic modes has been reported in $Bi_{1-x}Sb_x$[18] and bismuth[19], both of which are known to be weak TI candidates. Similar phenomena, if exist in superconductors (SC), would lead to a new mechanism of enabling Majorana modes. Indeed, previous theoretical studies have discussed this intriguing possibility of dislocation Majorana bound states (dMBSs) in $p$-wave topological superconductors[20-22]. However, due to the scarcity of realistic candi-date $p$-wave systems, we are not aware of any experimental progress along the search for dMBSs.

Recent years have also witnessed a Majorana revolution in the high-$T_c$ topological iron-based superconductors (tFeSCs), including $FeTe_{1-x}Se_x$[23-25], (Li,Fe)OHFeSe[26], LiFeAs[27-29], etc. Notably, the topology of tFeSCs only lies in their normal states[30], that a band inversion at the $Z$ point generates both a nontrivial $\mathbb{Z}_2$ electronic band topology and a helical Dirac surface state[31,32]. Below the critical temperature $T_c$, a nodeless pairing gap is developed for both bulk and surface states,

[1]Department of Physics and Astronomy, The University of Tennessee, Knoxville, TN, USA. [2]Institute for Advanced Materials and Manufacturing, The University of Tennessee, Knoxville, TN, USA. [3]Center for Correlated Matter and School of Physics, Zhejiang University, Hangzhou, China. [4]Department of Materials Science and Engineering, The University of Tennessee, Knoxville, TN, USA. ✉e-mail: ruixing@utk.edu

wiping out all normal-state topological physics around the Fermi energy. Despite the bulk-state triviality, striking evidence of Majorana signals has been extensively reported in superconducting vortices[23–29], atomic vacancies[33], and magnetic adatoms[34,35]. While the vortex Majorana signals in tFeSCs are usually believed to arise from the Fu-Kane mechanism[36–38], origins of the vacancy/impurity-related zero-bias peaks are still under debate[39–44]. Noting that vacancies and add-on impurities are both locally removable and point-like, one may also wonder if extended irremovable lattice defects such as dislocations or disclinations could invoke any interesting field-free topological physics in tFeSCs.

Our main finding in this work is that screw or edge dislocations can naturally bind 0D Majorana zero modes in tFeSCs and similar superconducting systems, in the absence of any external magnetic field. Noting that a pair of dislocations, as well as the 2D "cutting plane" attached to them, can be viewed as an effective 2D subsystem embedded in a 3D crystal, the four dislocation Majorana bound states manifest as "corner" Majorana modes for this 2D subsystem, one at each corner. Therefore, our mechanism exemplifies an unprecedented Majorana mechanism that is based on the second-order topology of 2D subsystems, which is in sharp contrast with earlier proposals on vortex/vacancy Majorana modes enabled by the first-order topology for 1D subsystems (i.e., vortex/vacancy lines). We thus dub this new phase "embedded second-order topological phase" ($ET_2$).

$ET_2$ in tFeSCs is completely driven by the normal-state topology[31,32], screw dislocations[45], and surface magnetism **M** that coexists with superconductivity[46–49], all of which have been experimentally observed in FeTe$_{1-x}$Se$_x$. In particular, we show that dMBSs emerge once the dislocation Burgers vector $\mathbf{b} = (b_x, b_y, b_z)$ satisfies $b_z \equiv 1 \bmod 2$, as a result of nested mass domains for surface Dirac fermions. Remarkably, this $ET_2$ condition is a natural outcome of a less recognized weak topological index $\boldsymbol{\nu} = (0, 0, 1)$ of tFeSCs. Therefore, our theory is directly applicable to other weak-index-carrying superconducting topological materials, such as $\beta$-Bi$_2$Pd[50]. We further discuss the impact of $s_\pm$-wave pairing symmetry on our recipe, and find it capable of inducing a new type of class DIII $ET_2$ with dislocation Majorana Kramers pairs (dMKPs), in the absence of any surface magnetism. Promising real-world material candidates and experimental signatures are also discussed.

## Results

### Dislocation Majorana bound states

We start by deriving the key result in our work, the recipe for dMBSs in tFeSCs, with the help of a nested domain wall approach. This construction scheme bears a resemblance to the "Dirac hierarchy" discussed in the previous literature[51–57]. We then proceed to discuss boundary conditions of dislocation-induced cutting plane and find that a 0D dMBS can be "inflated" to a 1D "hinge" chiral Majorana fermion under certain circumstances. Nonetheless, each corner of the cutting plane will always host a single zero-energy mode. This directly leads to the concept of embedded higher-order topological phase $ET_2$.

Let us first provide some motivations for our recipe. To trap a 0D bound state in a 3D system, one can start from a 3D gapless quasiparticle (e.g., massless Dirac fermion) and further constrain its degrees of freedom (d.o.f.) in all three spatial directions. This "dimensional reduction" procedure can be feasibly achieved by decorating the Dirac fermion with a hierarchical set of $\mathbb{Z}_2$ mass domains, with each domain effectively reducing the dimension of the gapless state by one. For example, the 2D gapless surface (i.e., a 2-fold Dirac fermion) of a 3D TI can be viewed as a domain wall-bound state for a 3D massive Dirac fermion, with the TI bulk and the outside vacuum carrying opposite Dirac masses, respectively. A second SC/magnetism domain for the 2D surface Dirac fermion further reduces the gapless d.o.f. to 1D, i.e., leading to a 1D chiral Majorana domain-wall mode[58,59]. To eventually achieve a 0D Majorana mode, it requires a third $\mathbb{Z}_2$-type mass domain

wall. We will show that, under certain circumstances, lattice domains introduced by screw/edge dislocations can serve as mass domains and thus contribute the last piece of the jigsaw puzzle. This approach is thus dubbed a nested domain wall construction for defect MBSs.

Another key motivation is from the material side. Recent experimental breakthroughs have revealed hidden topological Dirac surface states for several high-Tc iron-based SCs[31,32]. Among these tFeSC candidates, FeTe$_{1-x}$Se$_x$ is of particular interest to us, as it additionally harbors surface ferromagnetism that coexists with bulk superconductivity below its superconducting $T_c \sim$ 14.5 K[46–49]. Furthermore, screw dislocations for FeTe$_{1-x}$Se$_x$ can be generated in a highly controllable manner during the growth process[45]. Therefore, it is natural to expect FeTe$_{1-x}$Se$_x$ to be a wonderful playground for studying a new lattice topological defect-based Majorana platform in the absence of any external magnetic field. A possible recipe for Majorana bound states will be extremely helpful in diagnosing the topological situation here.

We now derive the topological condition of defect MBSs for tFeSCs. Our starting point is a 3D TRI TI with bulk isotropic $s$-wave spin-singlet superconductivity. The normal-state topology is indicated by a strong $\mathbb{Z}_2$ topological index $\nu_0$ and a set of weak $\mathbb{Z}_2$ indices $\boldsymbol{\nu} = (\nu_1, \nu_2, \nu_3)$[36]. In particular, $\nu_0 = 0$ ($\nu_0 = 1$) dictates an even (odd) number of Dirac surface states, while the values of weak indices $\nu_{1,2,3}$ decide the momentum-space locations of the surface states. The bulk $s$-wave SC, however, necessarily spoils the normal-state topology by introducing an isotropic SC gap $\delta_{\mathrm{SC}}$ to all Dirac surfaces through a "self-proximity" effect. Motivated by FeTe$_{1-x}$Se$_x$, we further introduce surface magnetism $\delta_{\mathrm{M}}$ to both the top and bottom (001) surfaces of our TI system. The explicit type of magnetism is flexible as long as it can act as a mass term for the Dirac surface state and further competes with the surface SC. Since the side surfaces are magnetism-free, when

$$|\delta_{\mathrm{M}}| > |\delta_{\mathrm{SC}}|, \tag{1}$$

a SC/magnetism domain emerges around the edges between top/bottom and side surfaces. This condition thus generates a 1D chiral Majorana mode around both top and bottom surfaces, i.e., a chiral Majorana hinge mode. We emphasize that the chiral Majorana hinge mode here is a result of 2D surface topology alone, that the top and bottom surfaces both feature a BdG Chern number of $|\mathcal{C}| = 1$. The 3D bulk topology will not be altered and thus remains trivial throughout the surface magnetism decoration.

Our last ingredient, the lattice dislocations, is intuitively a "gluing fault" when combining two identical copies of our setup. For example, as schematically shown in Fig. 1a, the screw dislocations are formed when the left parts of the two crystals are combined perfectly, while the right parts mismatch with each other by a displacement vector $\mathbf{b} = (0,0,1)$, i.e., the Burgers vector. While a screw or an edge dislocation appears one-dimensional, it must be attached to a 2D cutting plane $\mathcal{P}_c$ that only terminates at either another dislocation to form a dislocation dipole or the crystal boundary. An example of a cutting plane is highlighted by the orange line in Fig. 1a.

To explore the fate of chiral Majorana hinge modes during the gluing process, it is helpful to fold the top surfaces of the two to-be-glued crystals as shown in Fig. 1b. Then the previous interfacial problem is mapped to a 2D bilayer system in the $y$-$z$ plane, with each layer hosting a TI surface state. Distribution of $\delta_{\mathrm{M}}$ and $\delta_{\mathrm{SC}}$ are shown in Fig. 1c. The domain wall will bind a pair of counterpropagating chiral Majorana modes as denoted by the green and red arrows in Fig. 1c. Combining the two crystals is equivalent to introducing an interlayer coupling $t$ for only the bottom parts of the bilayer, i.e., the previous side surfaces, which will also couple the oppositely propagating Majorana modes and gap them out. However, the interlayer mass term for the Majorana fermions will obtain a phase factor $e^{i\pi\mathbf{b}\cdot\boldsymbol{\nu}}$, following the side Dirac surface states[3]. In the presence of a lattice dislocation, the

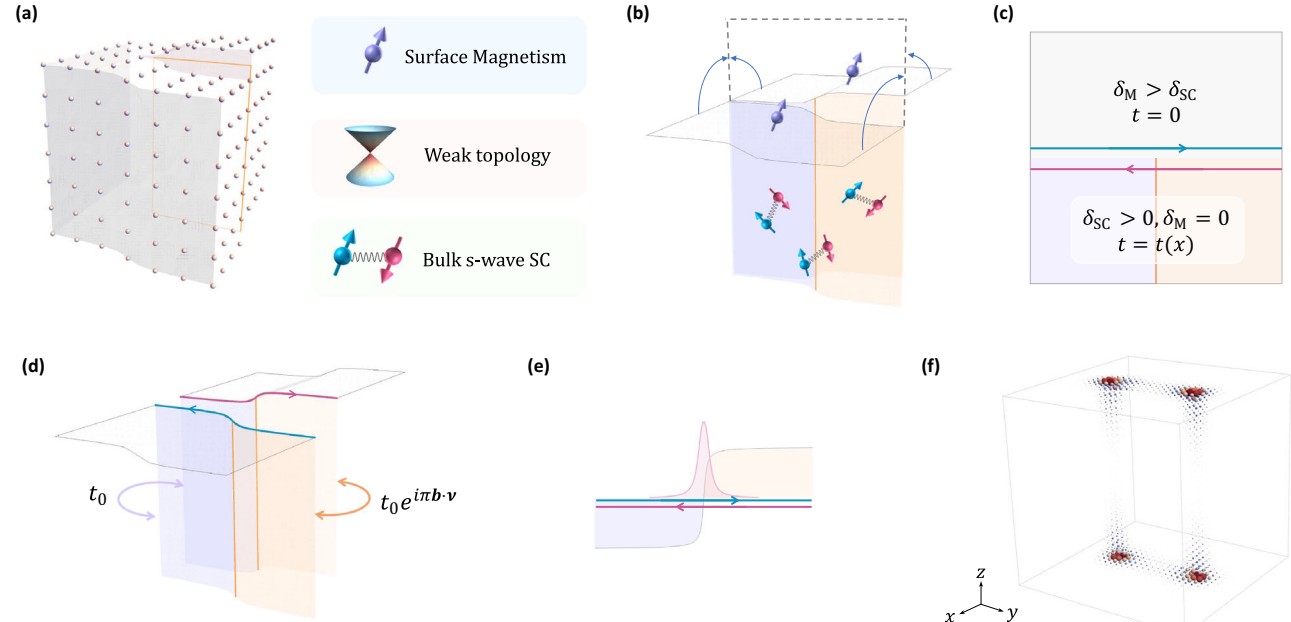

**Fig. 1 | Nested domain wall theory for the class-D embedded second-order topological phase ($ET_2$). a** A single screw dislocation with a Burgers vector **b** = (0, 0, 1) along with other key ingredients for $ET_2$: weak $\mathbb{Z}_2$ index in the normal state, bulk SC, and surface magnetism. In (**b**), we cut the crystal in halves following the orange cutting plane in (**a**), which leads to two disjoint magnetism-gapped top surfaces and two SC-gapped side surfaces. Further folding the top surfaces following the trajectory arrows leads to the "bilayer" configurations of Dirac surface states in (**c**). The competition between magnetism (M) and SC leads to a pair of counterpropagating 1D Majorana modes once $\delta_M > \delta_{SC}$. In (**d**), we glue everything together to restore the crystal, and the introduction of a dislocation decorates the intersurface hopping between Dirac particles on the orange cutting plane with a phase factor of $e^{i\pi \mathbf{b}\cdot\mathbf{v}}$. This gaps out the Majorana modes in a nontrivial way shown in (**e**), which can be mapped to a 1D Jackiw-Rebbi domain wall problem and results in a localized Majorana zero mode at the surface dislocation core. We carry out a numerical simulation of a pair of screw dislocations for $FeTe_{1-x}Se_x$ on a $28 \times 28 \times 28$ lattice. Four zero-energy modes are found and their spatial wavefunctions are found to be localized around each dislocation core, as shown in (**f**).

cutting plane [i.e., orange region in Fig. 1c] features a finite Burgers vector **b**, while the purple region has a zero Burgers vector because of the perfect lattice matching. Assuming the dislocation at $y = 0$, we have the mass term

$$t(y) = \begin{cases} t_0 & y < 0, \\ t_0 e^{i\pi\mathbf{b}\cdot\mathbf{v}} & y > 0. \end{cases} \quad (2)$$

Crucially, when

$$\mathbf{b} \cdot \mathbf{v} = 1 \bmod 2, \quad (3)$$

we have $t(y) = t_0 \mathrm{sgn}(y)$. Namely, when Eq. (3) is fulfilled, the chiral Majorana pair at the SC/magnetism domain experiences an additional mass domain due to the dislocation-induced lattice mismatch. This exactly resembles a 1D Jackiw-Rebbi problem and further results in a Majorana zero mode (MZM) localized around the defect core, as shown in Fig. 1e, completing the final part of our nested domain wall construction for defect MBSs. Similar nested domains will simultaneously show up for the dislocation core at the bottom surface and the other two corners of the cutting plane. This is how both Eq. (1) and Eq. (3) together serve as a sufficient topological condition for defect MZMs.

In Supplementary Note 1, we have developed an analytical theory for the dMBS, following the nested domain wall construction. In particular, we find that the in-plane localization length of the dMBS wavefunction yields a simple relation,

$$\xi_{MZM} \propto \frac{v_D}{|\delta_M| - \sqrt{\delta_{SC}^2 + \mu^2}}. \quad (4)$$

Here $v_D$ is the velocity of the Dirac fermion, and $\mu$ denotes the chemical potential for the Dirac point. Increasing either $v_D$ or $\delta_{SC}$ enhances $\xi_{MZM}$,

while the effect of $\delta_M$ is exactly the opposite. We have further compared the above analytical understanding with numerical studies of $\xi_{MZM}$ and an excellent agreement has been found.

## Boundary conditions & Majorana inflation

Geometrically, a dislocation-induced cutting plane $\mathcal{P}_c$ can terminate at either a crystal surface or another dislocation, leading to two seemingly different yet equivalent boundary conditions. For example, we can start from a dislocation dipole (i.e., a pair of dislocation lines) and move one dislocation towards the crystal side surface. This expands $\mathcal{P}_c$ until the dislocation hits the side surface and further merges with it. This process is reversible and thus transforms the aforementioned boundary conditions from one to another. Note that the shape of $\mathcal{P}_c$ could be variable in a realistic system since it can be viewed as a deformable membrane with a pair of fixed edges (i.e., the dislocations)[60]. Nonetheless, it is easy to see that the fate of dMBSs is solely determined by the conditions identified in the previous section, and is thus independent of the geometric details of $\mathcal{P}_c$.

In Fig. 2a, we schematically show the distributions of Majorana modes for the dislocation-dipole geometry. Each of the four dislocation cores will bind one MZM denoted by the quasiparticle operators $\gamma_i = \gamma_i^\dagger$ with $i \in \{1, 2, 3, 4\}$. In the cylindrical geometry shown in Fig. 2a, the chiral Majorana hinge modes always feature a finite-size gap that is inversely proportional to the cylinder radius[61], as shown in Fig. 2b. This gap is a manifestation of the anti-periodic boundary condition of 1D Majorana modes and can be removed by updating the boundary condition to a periodic one with a $\pi$-flux insertion. Thus, despite their chiral Majorana dispersions, the hinges do not carry any strictly zero-energy mode when they enclose a dislocation dipole.

Because of this finite-size hinge gap, when the defect MZM merges with the hinge Majorana modes as shown in Fig. 2c, its zero-energy nature remains. This is because a zero mode can only be spoiled while

interacting with another zero mode. Numerically, we find that the defect MZM eventually merges with the 1D chiral hinge mode, making the hinge harbor a 1D zero-energy state at $k = 0$, as schematically shown in Fig. 2d. Therefore, the corner-localized 0D MZMs of $ET_2$ can be inflated to 1D zero modes by simply changing the terminations of the cutting plane $\mathcal{P}_c$. Notably, this inflation process is reversible, and one can similarly "condense" a 1D zero mode into a 0D dMBS by recovering the dislocation dipole.

## Embedded higher-order topology

The fact that dMBSs are "corner" Majorana modes of the cutting plane motivates us to define a higher-order topology[62–66] for the dislocation-spanned subsystem. In particular,

**Definition 1.** *An embedded nth-order topology (dubbed $ET_n$) is defined by the presence of $(d − n)$-dimensional gapless boundary of a $d$-dimensional subsystem, which is further embedded in a $D$-dimensional bulk system with $D > d > n > 0$.*

Thus, $ET_n$ is a higher-order generalization of the "embedded topology" proposed in refs. 67,68. Our recipe for dMBSs features $D = 3$, $d = 2$, and $n = 2$, which thus corresponds to a class D $ET_2$ phase by definition.

Finally, it is instructive to review, clarify, and summarize the topological physics at each level of our dMBS recipe, which is illustrated in Table 1. First, we require the normal state of a target system to carry a nontrivial weak $\mathbb{Z}_2$ index, e.g., $\nu_z = 1$. In contrast, the bulk superconducting ground state is topologically trivial as a consequence of the spin-singlet $s$-wave pairing that is considered. When the surface magnetism kicks in, the superconducting surface states now carry a nontrivial 2D class-D topology, further leading to the dMBSs at the surface dislocation cores. Therefore, our theory for dMBSs is based on a trivial bulk SC with a nontrivial normal state and is thus distinct from previous proposals where the bulk TSC physics is a necessary ingredient.

## Model Hamiltonian

In this section, we provide a minimal lattice model for $FeTe_{1−x}Se_x$ to demonstrate the above $ET_2$ recipe. Bulk superconductivity and surface

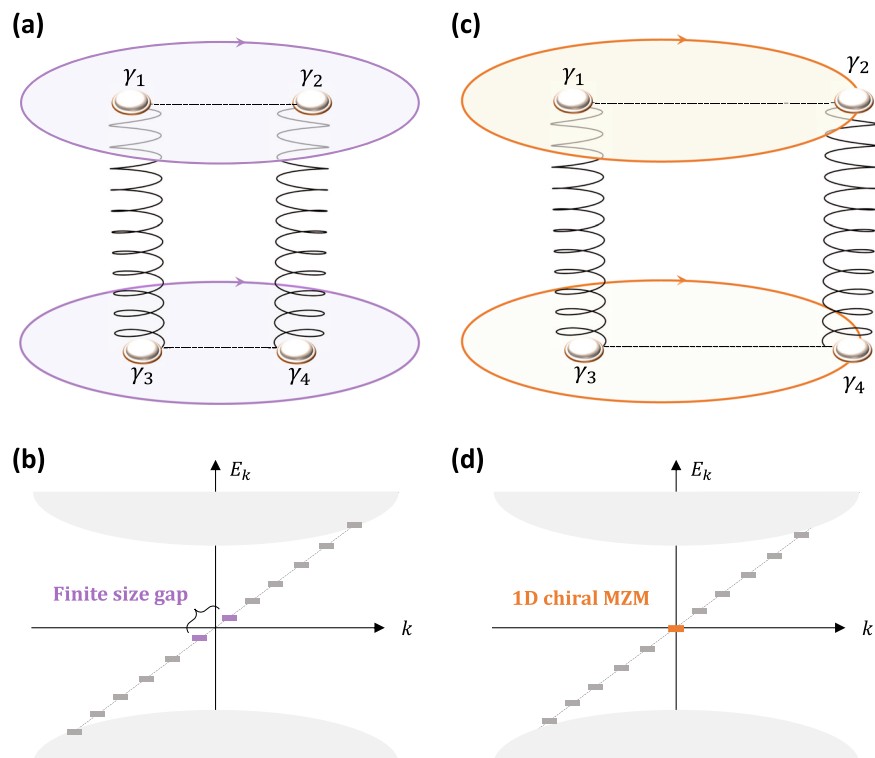

**Fig. 2 | "Inflation" of a Majorana mode from 0D to 1D. a** A schematic of a dislocation dipole and its associated dislocation Majorana bound states (dMBSs). The surface Chern number enforces a pair of 1D chiral Majorana modes circulating the top/bottom surface. The chiral Majorana modes yield a finite-size energy gap, as schematically shown in (**b**). When the cutting plane terminates at the sample boundary shown in (**c**), the two dMBSs $\gamma_{2,4}$ move to the hinges and merge with the chiral hinge modes, decorating each 1D chiral mode with a zero-energy state, as shown in (**d**). Notably, the total number of zero-energy Majorana modes remains to be four while evolving from (**a**) to (**c**).

**Table 1 | Summary of topological physics at each level of the dMBS theory in general $s$-wave superconductors**

|  | Normal state | Bulk SC states | Surface SC states | Dislocations |
|---|---|---|---|---|
| Dimension | 3D | 3D | 2D | 2D |
| Symmetry Class | All | DIII | D | D |
| Topology | weak index | trivial | BdG Chern number | $ET_2$ |
| Origins | band inversions | s-wave singlet pairing | Eq. (1) | Eq. (1) and Eq. (3) |
| Boundary Physics | Dirac surface states | none | chiral Majorana hinge modes | dMBSs |

*SC superconductor, $ET_2$ embedded second-order topological phase, dMBSs dislocation Majorana bound states.*

ferromagnetism (FM) are also included in our model setup. By analyzing the competition of SC and FM for the Dirac surface states, we map out a surface topological phase diagram to discuss when Eq. (1) will be fulfilled. This can be directly translated to a condition for $ET_2$ to emerge in FeTe$_{1-x}$Se$_x$, which we verify through explicit screw dislocation simulations for our minimal model.

Our minimal Bogoliubov-de Gennes (BdG) Hamiltonian for FeTe$_{1-x}$Se$_x$ is

$$\mathcal{H}_{BdG}(\mathbf{k}) = \begin{pmatrix} \mathcal{H}_0(\mathbf{k}) - \mu & \Delta(\mathbf{k}) \\ \Delta^\dagger(\mathbf{k}) & -\mathcal{H}_0^*(-\mathbf{k}) + \mu \end{pmatrix}, \tag{5}$$

where the normal-state Hamiltonian $\mathcal{H}_0 = v(\sin k_y \Gamma_1 - \sin k_x \Gamma_2 + \sin k_z \Gamma_4) + m(\mathbf{k})\Gamma_5$. The $\Gamma$ matrices are $\Gamma_1 = \sigma_x \otimes s_x$, $\Gamma_2 = \sigma_x \otimes s_y$, $\Gamma_3 = \sigma_x \otimes s_z$, $\Gamma_4 = \sigma_y \otimes s_0$, $\Gamma_5 = \sigma_z \otimes s_0$, where $s_{0,x,y,z}$ and $\sigma_{0,x,y,z}$ are Pauli matrices for spin and orbital d.o.f., respectively. Here $m(\mathbf{k}) = m_0 - m_1(\cos k_x + \cos k_y) - m_2 \cos k_z$ and $\mu$ is the chemical potential. We choose $v = 1, m_0 = -4, m_1 = -2, m_2 = 1$ to ensure a single topological band inversion at $Z$[69,70], leading to $\nu_0 = 1$ and $\mathbf{\nu} = (0,0,1)$. This well matches the low-energy topological band ordering of FeTe$_{1-x}$Se$_x$. To introduce superconductivity, we adopt a spin-singlet extended s-wave pairing for our model, where the pairing matrix $\Delta(\mathbf{k}) = [\Delta_0 + \Delta_1(\cos k_x + \cos k_y)](i\sigma_0 \otimes s_y)$. Here $\Delta_0$ ($\Delta_1$) is the on-site (nearest-neighbor) intra-orbital pairing strength.

Finally, following the experimental observations of FeTe$_{1-x}$Se$_x$ in refs. 46,47,49, we introduce uniform surface ferromagnetism to both top and bottom (001) surfaces in a finite-size slab geometry, with $N_z$ layers stacked along $\hat{z}$ direction. $\mathcal{H}_{FM} = f(z)[g_1\sigma_0 + g_2\sigma_z] \otimes (\mathbf{s} \cdot \mathbf{M})$ with $f(z) = \delta_{z,1} + \delta_{z,N_z}$ for a lattice layer index $z = 1,2,\ldots,N_z$. Here $\delta_{z,i}$ is the Kronecker delta function, $\mathbf{M}$ denotes the surface magnetization, and $g_1 \pm g_2$ are the effective isotropic Landé g-factor for the two orbitals involved in our model. We take $g_1 = 0.5$ and $g_2 = 0.2$ in our numerical simulations throughout this work. More discussions on the experimental aspects of FeTe$_{1-x}$Se$_x$ and other candidate materials will be presented later.

## Surface topological phase diagram: condition for dMBSs & partial fermi surface

The first step to realize dMBSs or class-D $ET_2$ is to identify the concrete condition to achieve Eq. (1) for our system by studying the competition between magnetism and superconductivity on the (001) surfaces. Note that the (001) Dirac surface state is localized around $\bar{\Gamma}$, the center of the surface Brillouin zone (BZ). As a result, the surface state will develop an isotropic pairing gap from the self-proximity effect[70], irrespective of the $s_\pm$ nature of $\Delta(\mathbf{k})$. The $s_\pm$ pairing will only play a role for $ET_2$ when the surface magnetism is absent (i.e., for symmetry class DIII), which will be discussed later. For FeTe$_{1-x}$Se$_x$ and its class D $ET_2$ physics, we can simplify the pairing term to an on-site s-wave type by setting $\Delta_1 = 0$.

We further remark that the surface Dirac fermion has a continuous rotation symmetry around the z-axis in the low-energy limit. Therefore, the effect of a general FM configuration $\mathbf{M} = (M_x, M_y, M_z)$ is always equivalent to that of $\mathbf{M}' = (0, M_\parallel, M_z)$ up to a coordinate transformation, where $M_\parallel = \sqrt{M_x^2 + M_y^2}$. Without loss of generality, we thus consider $\mathbf{M} = (0, M_y, M_z)$, and the surface Hamiltonian reads,

$$\mathcal{H}_{surf} = v_F(k_x \tau_z s_y - k_y \tau_0 s_x) - \mu \tau_z s_0 \\ + \Delta_0 \tau_y s_y + \Sigma_y \tau_0 s_y + \Sigma_z \tau_z s_z, \tag{6}$$

where $\mathbf{s}$ and $\mathbf{\tau}$ represent spin and particle-hole degree of freedom, respectively. $v_F$ is the surface Fermi velocity. Up to the first-order perturbation approximation, $\Sigma_y \approx g_1 M_y$ and $\Sigma_z \approx g_1 M_z$ are the Zeeman energies[69]. Notably, the condition of Eq. (1) is primarily concerned with the gap structures at $\bar{\Gamma}$. We thus find that $E_{\bar{\Gamma}} = \pm\sqrt{\mu^2 + \Delta_0^2} \pm \sqrt{\Sigma_y^2 + \Sigma_z^2}$. If we add back $\Sigma_x \approx g_1 M_x$, then the surface gap closing condition is

$\mu^2 + \Delta_0^2 = \mathbf{\Sigma}^2$ with $\mathbf{\Sigma} = (\Sigma_x, \Sigma_y, \Sigma_z)$. It is then easy to check that the $ET_2$ condition of Eq. (1) now becomes

$$|\mathbf{\Sigma}| > \sqrt{\mu^2 + \Delta_0^2}, \tag{7}$$

which coincides with the condition for the Dirac surface states to carry a BdG Chern number $|\mathcal{C}| = 1$. This nontrivial $\mathcal{C}$ accounts for the chiral Majorana hinge modes in Fig. 1c, a crucial step to complete the nested domain wall configuration for achieving $ET_2$. According to Eq. (3), a pair of screw or edge dislocations featuring an odd $b_z$ will span a 2D cutting plane with class D $ET_2$. Given the existence of such dislocations, Eq. (7) now serves as the $ET_2$ condition for FeTe$_{1-x}$Se$_x$.

On the other hand, a large in-plane $\mathbf{M}$ is capable of inducing partial Fermi surface (PFS) in a superconducting TI[71,72]. As shown in Fig. 3c, d, PFS occurs when some surface quasi-particle bands cross zero energy to form metal-like band patterns. While the formation of PFS is irrelevant to our target $ET_2$ physics, however, it can coexist with $ET_2$ and thus contributes an important part of our surface phase diagram. As an intuitive example, we consider $\mathbf{M} = (0, M_y, 0)$ and find the dispersion of $\mathcal{H}_{surf}$ at $k_y = 0$ is $E_{\alpha\beta}(k_x) = \alpha\Sigma_y + \beta\sqrt{(v_F k_x - \alpha\mu)^2 + \Delta_0^2}$ with $\alpha, \beta = \pm$. For $\alpha\beta < 0$, $E_{\alpha\beta}$ has two zero-energy solutions at $k_x = k_\alpha^{(\pm)}$, with

$$k_\alpha^{(\pm)} = \frac{1}{v_F}\left(\alpha\mu \pm \sqrt{\Sigma_y^2 - \Delta_0^2}\right). \tag{8}$$

Therefore, when $|\Sigma_y| \geq |\Delta_0|$, $E_{+-} = 0$ and $E_{-+} = 0$ lead to four $k_x$ solutions that form two sets of partial Fermi surfaces. Thanks to the rotation symmetry of Dirac surface Hamiltonian, we expect this PFS condition to be generalized to

$$|\Sigma_\parallel| \geq |\Delta_0| \quad \text{for } M_z = 0, \tag{9}$$

where $\Sigma_\parallel = g_1 M_\parallel$. Combining Eq. (7) with Eq. (9), we conclude that with $|M_z| \ll M_\parallel$, increasing $M_\parallel$ will always first drive the formation of PFS ($M_\parallel \approx |\Delta_0|$) before $ET_2$ phase is achieved ($M_\parallel = \sqrt{\Delta_0^2 + \mu^2}$).

The above analytical results are in excellent agreement with our numerical surface topological phase diagram in Fig. 3a. This $M_y$-$M_z$ phase diagram is essentially an energy-gap mapping of surface BdG spectrum for Eq. (5) in a thick slab geometry along $\hat{z}$ direction, where we take $\mu = \Delta_0 = 0.2$. The color in this logarithmic plot is a measure of the energy gap of the lowest BdG band, and in particular, regions colored in white feature a vanishing BdG gap, i.e., either a topological phase transition or a PFS phase. Our analytical condition of $ET_2$ (black dashed line) in Eq. (7) matches perfectly with the numerical finding in Fig. 3a. In addition, Eq. (9) predicts a critical $M_y^{(c)} = \Sigma_\parallel^{(c)}/g_1 = \Delta_0/g_1 = 0.4$, also agreeing with numerically-mapped boundary of PFS phase at $M_z = 0$. As shown in Fig. 3a, PFS survives until $M_z$ reaches a critical value of -0.4, and it is generally absent when $M_z > M_\parallel$. Importantly, PFS coexists with $ET_2$ most of the time in the phase diagram. So we expect that in a large $M_\parallel$ system, an observation of PFS will serve as a promising indicator for $ET_2$ in the system. However, it is also possible for the dMBS to interact with the gapless background of PFS, making it easier to hybridize with another Majorana mode at a neighboring dislocation.

In Fig. 3b, we further study the effect of chemical potential $\mu$ on the formation of $ET_2$. For a small $\mu$, the topological phase boundary separating $ET_2$ and the trivial phase is well captured by the dashed guideline predicted by Eq. (7). Notably, the phase boundary undergoes a sudden turn at $\mu \sim 0.7$ and starts to deviate from the analytical results. This is because the bulk-band physics is getting more involved as $\mu$ grows, and thus our effective surface theory is no longer expected to faithfully describe the phase boundary of $ET_2$.

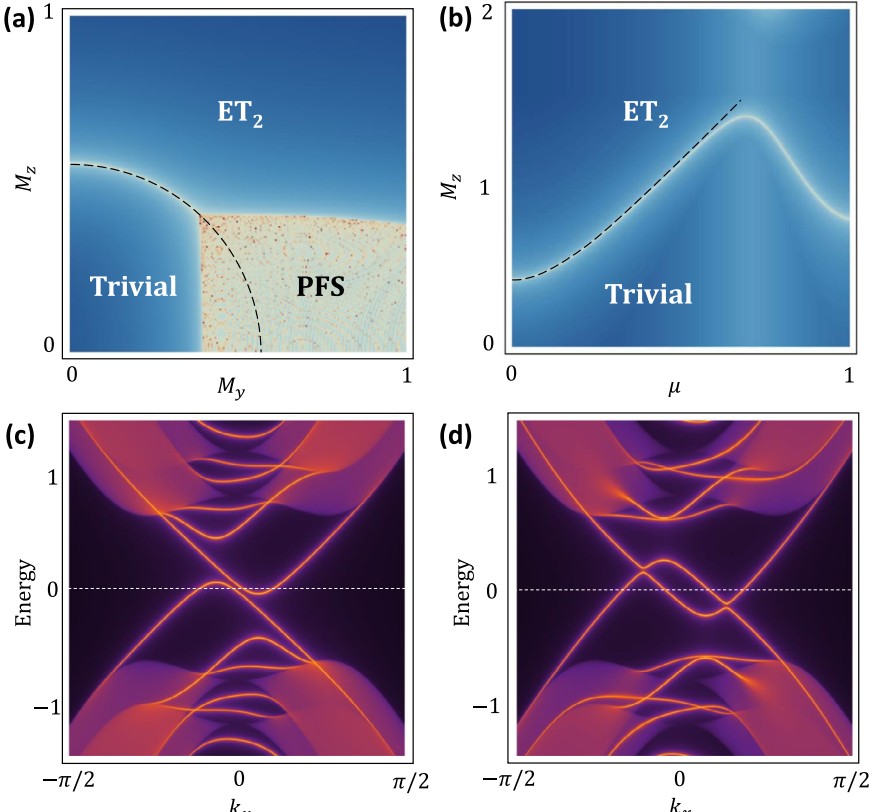

**Fig. 3 | Surface topological phase diagram and partial Fermi surface.** Surface topological phase diagrams as a function of $(M_y, M_z)$ and $(M_z, \mu)$ are shown in (**a**) and (**b**). The darker the blue color is, the larger the surface energy gap is. The region colored in white has no energy gap and could indicate the existence of either a surface topological phase transition or a partial Fermi surface (PFS). Note that embedded second-order topological phase ($ET_2$) and PFS can coexist. **c** The surface spectrum of a PFS along $k_x$ with $(M_y, M_z) = (0.5, 0.05)$. An example of the surface spectrum with coexisting $ET_2$ and PFS is shown in (**d**) with $(M_y, M_z) = (0.9, 0.05)$.

## Numerical simulation of dMBSs

To confirm the $ET_2$ phase, we consider to place our minimal model for FeTe$_{1-x}$Se$_x$ on a $28 \times 28 \times 28$ lattice. Periodic boundary conditions are considered for both $x$ and $y$ directions of the lattice cube to eliminate possible unwanted hinge modes in our simulations. Out-of-plane FM is considered for both the top and bottom layers of the lattice cube, following $\mathcal{H}_{FM}$. We further decorate our lattice system with a pair of screw dislocations with a Burgers vector $\mathbf{b} = (0, 0, 1)$. The dislocation dipole spans a 2D cutting plane $\mathcal{P}_c$ that is parallel to the $y$–$z$ plane. In principle, one can consider a pair of edge dislocations instead, as long as their Burgers vectors satisfy

$$b_z \equiv 1 \mod 2. \tag{10}$$

As the FM is gradually turned on, the (001) surface gaps close and reopen in our cubic geometry following Fig. 3a, after which four zero-energy modes show up in the energy spectrum. In Fig. 1f, we visualize the spatial distribution of the zero-mode wavefunctions in the cubic geometry and find each of the four surface dislocation cores is trapping one of the zero modes. These dislocation-bound Majorana zero modes are exactly the defining boundary signature of $ET_2$ in our system.

## $s_\pm$-wave pairing & class DIII $ET_2$

For tFeSC candidates such as FeTe$_{1-x}$Se$_x$ and LiFeAs, the bulk $s_\pm$ pairing as described by $\Delta(\mathbf{k})$ is supported by experimental observations[73–75]. In particular, $\Delta_1 \neq 0$ is crucial for enabling a relative $\pi$-phase difference for the local superconductivity orders of the $\Gamma$ and $M$ pockets. As we have discussed, in the above $ET_2$ recipe, it is the competition between SC and FM, rather than the explicit SC pairing type, that is crucial for

enabling the dislocation Majorana bound states. In this section, we show that $s_\pm$ pairing is indeed important for achieving a new class of time-reversal-invariant $ET_2$ in symmetry class DIII, but only when the surface FM is absent.

Our new recipe for class DIII $ET_2$ is motivated by the deep connection between hinge Majorana modes and $ET_2$, as revealed in the nested domain wall picture. Even in the absence of surface FM, a bulk $s_\pm$ pairing itself is capable of inducing a pairing mass domain for Dirac fermions living on the top (bottom) and side surfaces. As a result, the inter-surface hinge will harbor a pair of 1D helical Majorana modes that respect time-reversal symmetry[70]. As shown in Fig. 4, we can now follow a "cut and glue" procedure to reveal the dislocation physics. Cutting the crystal now yields two pairs of helical Majorana modes trapped to the top hinges of the two smaller crystals [as shown in Fig. 4a, b], as well as another two pairs bound to the bottom hinges. When gluing the crystal back together, a dislocation will introduce a $\pi$-phase domain to the inter-hinge binding term following Eq. (2), which will now trap a Kramers pair of Majorana zero modes around each of the surface dislocation core.

We now provide a lattice simulation to verify the existence of class DIII $ET_2$ with our minimal model of FeTe$_{1-x}$Se$_x$ in Eq. (5). We adopt the same model parameters of Fig. 1f, with no surface FM assumed and an additional update of $\Delta_0 = -0.85$ and $\Delta_1 = 0.5$ to emphasize the effect of $s_\pm$ pairing. Note that the $s_\pm$ condition for tFeSCs with $\Delta(\Gamma)\Delta(M) < 0$ is generally achieved when $|\Delta_0| < 2|\Delta_1|$. The energy spectrum for the system is calculated for a $36 \times 36 \times 20$ lattice geometry, with a pair of screw dislocations placed in the $y$-$z$ plane. As shown in Fig. 4c, eight Majorana modes (orange circles) show up in the energy spectrum that are well separated from other higher-energy states. The small energy splitting for the Majorana modes is due to the finite-size effect of the

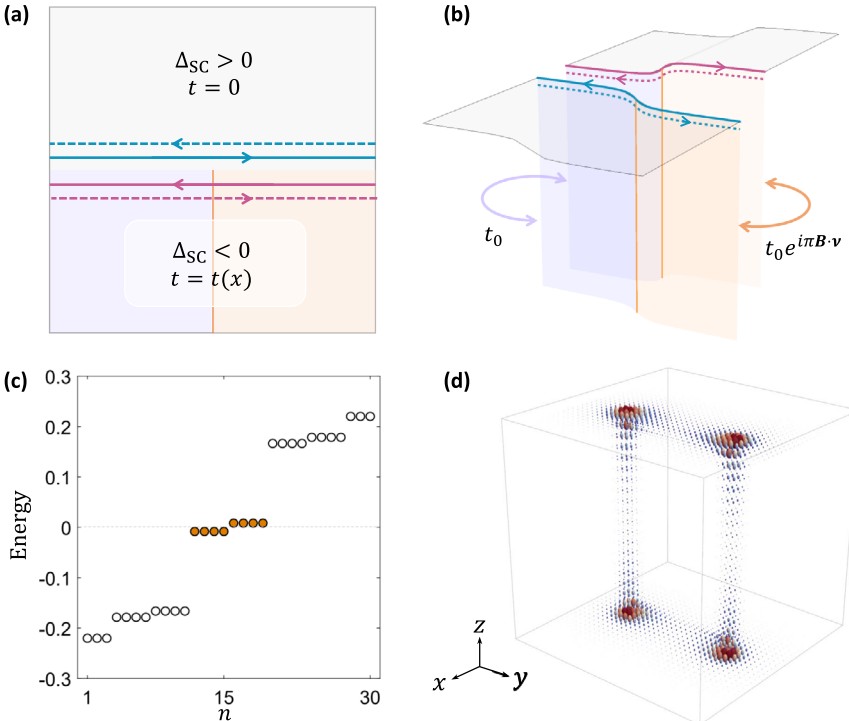

**Fig. 4 | Time-reversal-invariant embedded second-order topological phase (ET$_2$) driven by an extended $s$-wave pairing with $(\Delta_0, \Delta_1)$ = (−0.85, 0.5). a** and **b** illustrate a nested domain wall construction similar to that in Fig. 1. Two pairs of helical Majorana modes now show up, the gluing of which leads to a Kramers pair of Majorana bound states at each dislocation core. **c** The energy spectrum of a

$36 \times 36 \times 20$ lattice with a dislocation dipole that supports eight Majorana zero modes. By plotting the spatial wavefunction distribution of these Majorana modes in (**d**), we numerically confirm that each dislocation core binds one Majorana Kramers pair.

cubic geometry. By plotting the Majorana wavefunctions in the real space, we find in Fig. 4d that each surface dislocation core now harbors a pair of Majorana modes, which unambiguously demonstrates the existence of class DIII ET$_2$ trapped by the lattice dislocations.

## Material candidates

In this section, we will discuss material candidates that can harbor ET$_2$ physics in both class D and class DIII. We will focus on the tFeSCs, especially FeTe$_{1-x}$Se$_x$ and LiFeAs, and further discuss their experimental relevance. However, ET$_2$ is not a privilege of tFeSCs and can in principle exist in other superconducting systems as well. We will discuss $\beta$-PdBi$_2$ as such an example. A brief summary of candidate systems can be found in Table 2.

**FeTe$_{1-x}$Se$_x$.** As discussed above, FeTe$_{1-x}$Se$_x$ naturally combines all necessary ingredients of our class D ET$_2$ recipe and manifests itself as perhaps the most promising platform for dMBSs. Thanks to the recent extensive experimental studies on both the normal-state topology and high-temperature superconductivity of FeTe$_{1-x}$Se$_x$[30–32], we are capable of discussing its ET$_2$ possibility quantitatively.

Evidence of surface magnetism in FeTe$_{1-x}$Se$_x$ has been experimentally established by a variety of measurement approaches, as summarized in Table 3. For example, an angle-resolved photoemission spectroscopy (ARPES) study in ref. 46 reveals a direct surface gap of ~8 meV exactly at the surface Dirac point, in addition to the surface SC gap at the Fermi level. The spoiling of the Kramers degeneracy of the Dirac surface state happens even above the superconducting transition temperature $T_c$, directly implying the breaking of time-reversal symmetry. Even though other more complex scenarios such as time-reversal-broken superconductivity is in principle possible[76,77], a most straightforward interpretation of this magnetic gap would be the development of out-of-plane FM order on the surface. Similar evidence

of surface FM has also been detected by the nanoscale quantum sensing of magnetic flux by nitrogen vacancy (NV) centers[47], where the magnetization is reported to feature an in-plane component as well. In a recent transport measurement, a coexistence of in-plane magnetization and superconductivity has also been observed in van der Waals Josephson junctions fabricated with Fe(Te,Se)[78].

Earlier experimental studies[31] further reveal a surface superconducting order of $\Delta_0 \sim 2$ meV and a chemical potential of $\mu \sim 4.4$ meV, in addition to $\Sigma_z \sim 4$ meV. Considering the condition in Eq. (7), ET$_2$ phase can be achieved with either (i) a slight electron doping to reduce $\mu$, or (ii) an enhancement of surface FM. Notably, engineering surface FM could be more experimentally accessible. For example, neutron

**Table 2 | Candidate materials for embedded second-order topological phases (ET$_2$)**

| Materials | SCing $T_c$ | $\mathbb{Z}_2$ Index | Bound State |
|---|---|---|---|
| FeTe$_{0.55}$Se$_{0.45}$ | 14.5 K | (1; 0, 0, 1) | MZM |
| LiFeAs | 17 K | (1; 0, 0, 1) | MKP |
| (Li,Fe)OHFeSe | 41 K | (1; 0, 0, 0) | N/A |
| $\beta$-PdBi$_2$ | 5.3 K | (1; 0, 0, 1) | MZM |

Candidates with dislocation Majorana zero modes (MZMs) or Majorana Kramers pairs (MKPs) can realize an ET$_2$ of class D or DIII. (Li,Fe)OHFeSe is not expected to carry any ET$_2$ physics.

**Table 3 | Summary of experiments on the surface magnetism in FeTe$_{1-x}$Se$_x$ with ARPES[46,48] and NV center[47]**

| Probe | Mag. type | Orientation | Surf. Gap |
|---|---|---|---|
| ARPES | FM | $\hat{z}$ | ~ 8 meV |
| NV Center | FM | $\hat{x}$-$\hat{z}$ | N/A |

scattering measurements have revealed that a single interstitial Fe impurity can induce magnetic Friedel-like oscillation involving > 50 neighboring Fe sites[79]. As a result, an interstitial Fe impurity on the surface is capable of generating a local magnetic patch with $\Sigma_z \sim 10$ meV[80], which is large enough to enable $ET_2$. Although interstitial Fe impurities could naturally exist during sample growth, they can also be deposited to the sample surface as adatoms[81]. This provides us with a highly controlled approach to enhance the surface FM of general tFeSCs.

Remarkably, for $FeTe_{1-x}Se_x$ films epitaxially grown with pulsed laser deposition (PLD), the formation of screw dislocations can be feasibly controlled by simply tuning the deposition rate[45]. In particular, samples grown at a low deposition rate generally feature spiral-like surface morphology that encodes a screw dislocation with a Burgers vector of $\mathbf{b} = (0, 0, 1)$. This thus contributes the last key ingredient for materializing dMBSs in $FeTe_{1-x}Se_x$ at zero magnetic fields.

In Supplementary Note 2, we have numerically explored the $ET_2$ condition for $FeTe_{0.55}Se_{0.45}$ based on an eight-band $\mathbf{k} \cdot \mathbf{p}$ model[37] that reproduces the low-energy band structures of first-principles calculations. Remarkably, the surface topological phase diagram based on this realistic model is in excellent agreement with Fig. 3b. This not only proves the power of our minimal model approach, but also offers quantitative guidance for the experimental search of both dMBS and PFS.

**Other Fe-based superconductors.** Besides $FeTe_{1-x}Se_x$, evidences of Dirac surface states and vortex Majorana modes have also been found in other tFeSCs such as $LiFeAs$[27–29] and $(Li,Fe)OHFeSe$[26]. We first note that the topological band physics in $(Li,Fe)OHFeSe$ is mainly attributed to the band inversion at $\Gamma$, thus leaving the system with zero weak indices. We therefore do not expect $(Li,Fe)OHFeSe$ to carry $ET_2$ physics proposed in this work. A similar conclusion could be reached for $CaKFe_4As_4$, whose normal-state band inversion also happens at $\Gamma$ due to a band folding effect[82].

The band structure of $LiFeAs$ resembles that of $FeTe_{1-x}Se_x$ and features a weak-index vector $\mathbf{\nu} = (0, 0, 1)$. While we are not aware of any surface magnetism for $LiFeAs$, evidence of $s_\pm$ pairing has been reported in earlier ARPES measurements[83]. This would make $LiFeAs$ a good platform to host class DIII $ET_2$ and the associated defect Majorana Kramers pairs.

**$\beta$-PdBi$_2$, bismuth, and beyond.** Just like $FeTe_{1-x}Se_x$, $\beta$-$PdBi_2$[50] features both a single band inversion at $Z$ and intrinsic SC with a transition temperature of $T_c = 5.3$ K. By evaporating Cr atoms on Bi-terminated surface of $\beta$-$PdBi_2$, scanning tunneling microscopy (STM) technique can organize Cr atoms into a magnetic lattice that competes with SC on the surface[84]. In particular, both FM and anti-FM can be achieved by simply adjusting the lattice constant of the Cr adatoms. Therefore, we expect our $ET_2$ results on $FeTe_{1-x}Se_x$ to be directly applicable to $\beta$-$PdBi_2$ as well.

$ET_2$ can also be achieved in an extrinsic manner by assembling all the necessary elements in a heterostructure. For example, candidates of weak topological insulators carrying $\mathbb{Z}_2$ indices $(\nu_0; \nu_1, \nu_2, \nu_3) = (0; 0, 0, 1)$ have been experimentally established in a plethora of Bi-related materials, including $BiTe$[85], $Bi_2TeI$[86], $Bi_4I_4$[87], and $ZrTe_5$[88] etc. While these candidates are non-superconducting, one can design an ABC "trilayer" structure by growing a thin film of the above weak TIs on some superconducting substrates and further depositing another ferromagnetic layer on top. When a lattice screw dislocation with $\mathbf{b} = (0, 0, 1)$ occurs, the dMBSs should appear when Eq. (1) is satisfied.

Interestingly, the dMBSs for weak TIs with a trivial $\nu_0 = 0$ can be interpreted without exploiting the nested domain wall picture. This is because the dislocations-spanned cutting plane in a 3D weak TI with $\mathbf{\nu} = (0, 0, 1)$ effectively hosts an "embedded" quantum spin Hall (QSH)

phase, as experimentally confirmed in ref. 89. Namely, there exists a closed loop of 1D gapless helical electrons circulating the boundary of the cutting plane, thanks to the fact that the (001) surfaces are gapped. Should bulk SC and surface FM be simultaneously present, every two neighboring edges of this embedded QSH will be gapped differently, leading to "corner" MZMs[90]. However, this neat picture breaks down when $\nu_0 \neq 0$ and the (001) surface becomes gapless, as for the case in FTS. Notably, our theory of nested domain wall holds independent of the value of $\nu_0$, and it thus offers a more generalized perspective to comprehend the origin of dMBSs.

We further note that a similar structure has been successfully fabricated for Bi(111) grown on an Nb(110) substrate, of which a ferromagnetic Fe cluster is placed on top[91]. Notably, the topological nature of Bi is disputable because of the tiny energy gap at $L$ point, and Bi is believed to be either a higher-order topological insulator with trivial $\mathbb{Z}_2$ indices or a strong topological insulator with $(\nu_0, \nu_1, \nu_2, \nu_3) = (1; 1, 1, 1)$. Interestingly, the latter scenario is recently supported by the observation of helical electron modes bound to a screw dislocation via an STM study[19]. These experimental progresses have together established Bi as another promising platform for dMBSs.

**Experimental detection**

Signatures of $ET_2$ for the above material candidates can be feasibly revealed by mapping out the local density of states (LDOS) around lattice dislocations in experiments with the state-of-the-art STM technique. In this section, we numerically simulate the LDOS signals of dislocation-trapped Majorana modes for our minimal Hamiltonian in Eq. (5) using the iterative Green function method[92]. The geometry we considered involves an in-plane $20 \times 40$ lattice with a pair of screw dislocations embedded in the $y$–$z$ place, sitting symmetrically around the $z$-axis at $(x, y) = (10, 20)$. The spatial distance of the dislocations is denoted as $\delta r_d$. After sufficient iteration steps, the LDOS on the top (001) surface is $\mathcal{D}(\mathbf{r}, E) = -\frac{1}{\pi} \mathrm{Im} \left[ \mathcal{G}_{\mathrm{surf}}(\mathbf{r}, E) \right]$, where $\mathcal{G}_{\mathrm{surf}}(\mathbf{r}, E)$ is the surface Green function. This simulated LDOS signal can be directly compared with ultra-low-temperature STM data in future experiments.

When $\delta r_d$ is much greater than the Majorana localization length $l_M \sim 3$, the hybridization between neighboring defect Majorana modes is negligible, as shown in Fig. 5a. We then expect each dislocation to carry a sharp LDOS peak at the zero bias, as numerically confirmed in Fig. 5b. Moving away from the dislocation core, the peak intensity gradually drops to zero without any further splitting, implying the existence of a single zero-energy mode. Meanwhile, we carry out a similar simulation with $\delta r_d \sim \mathcal{O}(l_M)$, where the dMBSs hybridize strongly [Fig. 5c]. We similarly check the LDOS data near the bottom dislocation core and find the absence of any zero-bias peak in Fig. 5d. Instead, a double-peak structure emerges, indicating the annihilation of the dMBSs. As for $FeTe_{1-x}Se_x$, we expect $l_M$ to be of the order of the superconducting coherence length $\xi_{SC} \sim 5$ nm, similar to that of vortex Majorana modes[23]. This sets a crucial length scale for $\delta r_d$, that only when $\delta r_d \gg \xi_{SC}$ will a clear experimental Majorana signal be expected.

We now remark on several phenomenological distinctions between defect and vortex Majorana modes. First, a quantum vortex always traps finite-energy Caroli-de Gennes-Matricon (CdGM) states inside the SC gap, which can introduce Majorana-like signals near the zero energy and further complicate interpretations of experimental data. As for $ET_2$, however, we do expect the dislocation core to carry fewer or even no subgap states besides the dMBS, as shown in our numerical simulations. This "cleanliness" of the zero-bias signal of $ET_2$ is ascribed to the Jackiw-Rebbi nature of dMBS, which can significantly enhance the unambiguity of future experiments on relevant topics.

In the weak-pairing limit, the spatial distribution of a Majorana wavefunction should inherit the symmetry pattern of the local Hamiltonian in the normal state. Since a quantum vortex is usually rotational invariant, we thus expect the wavefunction of a vortex MBS to be circularly symmetric in general[11,93], unless an extra symmetry-

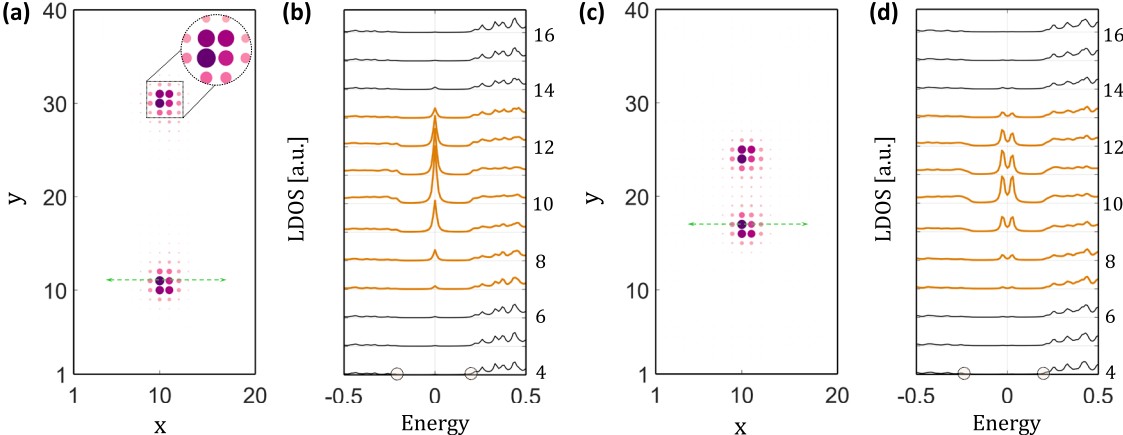

**Fig. 5 | Simulated surface local density of states (LDOS) patterns of dislocation Majorana bound states (dMBSs). a** The zero-energy top surface LDOS of two dMBSs that are far away, with each dislocation core trapping a well-defined zero mode. The inset is a zoom-in plot around the dislocation core, which clearly shows the spatial anisotropy of the Majorana wavefunction. By doing a line cut across the dislocation following the green reference line, energy-dependent LDOS plots at each site are shown in (**b**), which shows a sharp zero-energy LDOS peak. The two circles on the horizontal axis indicate the position of the surface magnetic gap. When the two dislocations are brought together in (**c**), the zero-energy LDOS peak splits due to Majorana hybridizations, as shown in (**d**).

breaking effect such as uniaxial strain or vortex-line tilting occurs. On the other hand, the LDOS profile of a dMBS is naturally anisotropic, since the geometry of a screw dislocation explicitly breaks the in-plane mirror symmetry of the underlying lattice, agreeing with our numerical simulations in Fig. 5. This pattern of dMBSs should be accessible via STM measurements.

Because of the known inhomogeneity of $FeTe_{1-x}Se_x$ samples, it is possible that local magnetic patches, instead of a uniform FM order, will appear on a real-world sample surface[48]. Motivated by this fact, we numerically test the fate of dMBSs in the presence of spatially fluctuating magnetic configurations. In particular, we couple the uniform ferromagnetic order $M_z$ with a spatially random perturbation $\widetilde{M}(\mathbf{r}) \in [-\Delta_M, \Delta_M]$. Upon disorder averaging, dMBSs are found to be extremely resilient against moderately strong magnetic disorders, especially when $\Delta_M < 3M_z$. We provide a detailed discussion of disorder effects in Supplementary Note 3.

We further notice that applying a $\hat{z}$-directional magnetic field can facilitate the creation of $ET_2$ phase by enhancing the magnetic gap, at the price of introducing additional SC vortex physics. While our dMBS is immune to an applied magnetic field, ref. 80 predicts that the field-induced SC vortices living inside the magnetic patch do not harbor any vortex Majorana modes, and are thus dubbed "empty vortices". With such an external magnetic field, we thus expect the dMBS to contribute the only zero-bias peak signal to an STM scanning inside a local FM patch, and it will be further surrounded by a set of "satellite" empty vortices with no Majorana signal. This unique phenomenon, if observed, will serve as rather compelling experimental evidence for $ET_2$.

## Discussion

In summary, we have proposed a new magnetic-field-free mechanism to trap non-Abelian Majorana zero modes with lattice dislocations in 3D $s$-wave superconductors with a trivial bulk-state topology at the BdG level. We further establish iron-based superconductors such as $FeTe_{1-x}Se_x$ as an ideal venue to realize dMBSs. This exotic defect Majorana physics manifests as an exemplar of an embedded higher-order topology, paving the way for exploring emergent subsystem topological physics. Notably, our recipe for dMBS is beyond tFeSCs and it further provides theoretical guidance to experimentally design and achieve dMBSs in other weak-index-carrying material systems such as $\beta$-PdBi$_2$. Given the remarkable capabilities of manipulating both the screw dislocations and surface

magnetism in $FeTe_{1-x}Se_x$ that have been reported in the literature, we believe that our proposal of dislocation Majorana physics will soon be experimentally realizable.

We further note that the $ET_2$ uncovered in this work is "extrinsic", in the sense that the dMBS cannot be characterized by a 2D bulk topological invariant of the subsystem. An intrinsic $ET_2$ phase should be symmetry-protected and is robust against any perturbations that do not close the local gap in the cutting plane. Recently, a relevant discussion in non-superconducting systems has been reported[94], where an inversion-protected $ET_2$ phase occurs as a response to the bulk higher-order topology. Given that TSCs are scarce in nature, it is thus highly desirable to explore whether intrinsic $ET_2$ phase or symmetry-protected dMBSs can emerge in a class-D or class-DIII topologically trivial superconductor. We leave this intriguing direction for future works.

## Methods

Here we present the analytical derivation of the localization length $\xi_{MZM}$ of 0D dislocation Majorana bound states. This is presented in Eq. (4), namely, $\xi_{MZM} \propto v_D/(|\delta_M| - \sqrt{\delta_{SC}^2 + \mu^2})$. For the detailed calculations, please refer to the Supplementary Note (1. A). We only outline the key steps here.

*Step 1*—For the topological surface state, we obtain the localized mode near $z = 0$ boundary between sample ($z > = 0$) and vacuum ($z < 0$)

$$\psi^\uparrow(z) = 2ce^{-Az}\sin(Bz)|\phi_-\rangle \otimes |\uparrow\rangle, \tag{11a}$$

$$\psi^\downarrow(z) = 2ce^{-Az}\sin(Bz)|\phi_-\rangle \otimes |\downarrow\rangle. \tag{11b}$$

where $|c| = \sqrt{A(A^2 + B^2)/(2A^2 + B^2)}$ is the normalization factor with $A = v_D/(2m_2)$ and $B = \sqrt{2m_{0z}m_2 - v^2}$. The spinor part provides the basis for Dirac surface states, $\{|\phi_-\rangle \otimes |\uparrow\rangle, |\phi_-\rangle \otimes |\downarrow\rangle\}$, where $\sigma_x|\phi_\pm\rangle = \pm|\phi_\pm\rangle$ leads to $|\phi_+\rangle = (1,1)^T/\sqrt{2}$ and $|\phi_-\rangle = (1,-1)^T/\sqrt{2}$. Thus, the surface state Hamiltonian up to linear k order,

$$\mathcal{H}_{surf} = v_D(k_x s_y - k_y s_x). \tag{12}$$

Here **s** are Pauli matrices for the spin degree of freedom. The localization length for the surface state is given by $\xi_{surf} = 1/A = 2m_2/v_D$, which indicates that a smaller spin-orbit coupling ($v_D$) corresponds to a larger localization length. In Supplementary

Note 2. A, we numerically find that the localization length of the Dirac surface state in FeTe$_{0.5}$Se$_{0.5}$ is about 30 layers (-18 nm). This localization length is much larger than the typical scale observed in Bi$_2$Se$_3$ and Bi$_2$Te$_3$ (-3 nm), as the spin-orbit coupling strength in FeTe$_{0.5}$Se$_{0.5}$ is smaller. Therefore, the FeTe$_{0.5}$Se$_{0.5}$ sample should be at least 60 layers in thickness along (001) direction, in order to observe the predicted dislocation MBS signals.

*Step 2*—On the surface of the sample, ferromagnetism coexists with bulk superconductivity in FeTe$_{0.5}$Se$_{0.5}$. Therefore, the BdG Hamiltonian for the top surface of the sample can be expressed as

$$\mathcal{H}_{BdG} = v_D(k_y s_y \gamma_z - k_x s_x \gamma_0) - \mu s_0 \gamma_z + \delta_M s_z \gamma_z + \delta_{SC} s_y \gamma_y, \tag{13}$$

where $\boldsymbol{\gamma}$ are Pauli matrices for the particle-hole degree of freedom. Without loss of generality, we consider $\delta_M > 0$ and $\delta_{SC}$. The gap closing occurs at $\mu^2 + \delta_{SC}^2 = \delta_M^2$, and topological gap is given by $\Delta_{topo} = \delta_M - \sqrt{\mu^2 + \delta_{SC}}$. For the topological phase of the sample surface, the boundary zero-mode solution for $\mathcal{H}_{BdG}(k_x = 0, -i\partial_y)$ is

$$\psi(y) = a e^{(-\delta_M + \delta_{SC})y/v_D} |\phi_+\rangle, \tag{14}$$

where $a$ is the normalization factor and the spinor part is $|\phi_+\rangle = (1, -i, -i, 1)^T/2$. The eigenstate $|\phi_+\rangle$ of the chiral symmetry $s_y\gamma_z$ satisfies $s_y\gamma_z|\phi_+\rangle = -|\phi_+\rangle$, and it is also an eigenstate of the particle-hole symmetry $s_0\gamma_x K$ with $s_0\gamma_x K|\phi_+\rangle = i|\phi_+\rangle$. Additionally, $|\phi_+\rangle$ is an eigenstate of the $k_x$ term, which leads to the 1D chiral Majorana mode dispersion along the 1D "hinge" of the sample surface. In particular, $\langle\phi_+|k_x s_y\gamma_z|\phi_+\rangle = k_x$.

*Step 3*—We next solve the dislocation MZM by introducing a pair of dislocation lines into the system. As discussed in Fig. 1 of the main text, the dislocation lines are oriented along the z-direction. The process involves two key steps.

- Cut the sample into two parts by the cutting plane expanded by this pair of dislocation lines, as illustrated in Supplementary Fig. 1c [see the dashed rectangle]. The 1D chiral Majorana mode is also divided into two parts, and near the touching edges, those two 1D chiral modes propagate along different directions due to the $C_{2z}$ symmetry [see Supplementary Fig. 1d].
- Glue two 1D chiral modes by restoring the lattice, as illustrated in Supplementary Fig. 1e.

As a result, we can construct an efficient two-by-two Hamiltonian that consists of two 1D chiral modes propagating in opposite directions, which are

$$\psi_R(y) \propto e^{(-\delta_M + \delta_{SC})y/v_D}(1, -i, -i, 1)^T, \tag{15a}$$

$$\psi_L(y) \propto e^{(\delta_M - \delta_{SC})y/v_D}(1, i, i, 1)^T. \tag{15b}$$

This gives rise to

$$\mathcal{H}_{dis} = v_D k_x \tau_z + \text{Im}[t_c]\tau_y + \text{Re}[t_c]\tau_x, \tag{16}$$

where the inter-edge coupling $t_c$ is due to the direct hopping $(v_D k_y s_x \gamma_0 \to -iv_D\partial_y s_x\gamma_0)$,

$$t_c \approx \int_{-\Delta_y}^{\Delta_y} dy \langle\phi_R|s_x\gamma_0|\phi_L\rangle \times \left(e^{(-\delta_M + \delta_{SC})y/v_D}\right.$$
$$\left.[-iv_D\partial_y]e^{(\delta_M - \delta_{SC})y/v_D}\right) \propto -iv_D\frac{\delta_M - \delta_{SC}}{v_D} \tag{17}$$
$$= -i(\delta_M - \delta_{SC}).$$

Please also note that, in the absence of dislocation pairs, $t_c$ is a constant; however, its sign varies depending on the position in the case of a dislocation[3]. For instance, these two dislocation lines are separated in real space and are located at $(N_{cx}, N_{cy}, z = 1) \to (N_{cx}, N_{cy}, z = N_z)$ [line 1] and $(N_{cx} + \Delta_x, N_{cy}, z = 1) \to (N_{cx} + \Delta_x, N_{cy}, z = N_z)$ [line 2]. The in-plane distance between these two dislocation lines is denoted by $\Delta_x$. Subsequently, we obtain

$$t_c = \begin{cases} -i(M_z - \Delta_0) \text{ for } x \leq N_{cx} \text{ or } x \geq N_{cx} + \Delta_x, \\ i(M_z - \Delta_0) \text{ for } N_{cx} \geq x \leq N_{cx} + \Delta_x. \end{cases} \tag{18}$$

The hopping term for spin-orbit coupling acquires a $\pi$ phase accumulation when circling the dislocation, which accounts for the minus sign. Consequently, $t_c$ serves as the mass term for the two 1D chiral Majorana modes, and its sign reverses. This leads to the formation of a 1D domain wall along the line connecting the two dislocation lines. A 0D MZM, referred to as the dislocation MBS in the main text, naturally emerges.

## Data availability
The datasets generated during this study are available upon request.

## Code availability
The custom codes generated during this study are available upon request.

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

## Acknowledgements

We thank X. Liu, C.-Z. Chen, X.-Q. Sun, and J. Yu for helpful discussions. We are particularly indebted to L.-Y. Kong and X. Wu for the inspiring discussions on the candidate materials. This work is supported by a start-up funding at the University of Tennessee.

## Author contributions

R.-X.Z. conceived the original idea and supervised the whole project. L.-H.H. performed the theoretical analysis and the numerical simulations with the help of R.-X.Z. Both authors contributed essentially to the manuscript preparation.

## Competing interests

The authors declare no competing interests.
