## [Peer Review File · Nature Communications]

REVIEWER COMMENTS

Reviewer #1 (Remarks to the Author):

In this work, the authors demonstrate that under certain conditions, a lattice dislocation in a topological superconductor will bind OD Majorana states. The authors support their findings with theoretical arguments, tight-binding calculations, and material analysis. The material realization discussion in particular is quite thoughtful, and the highlighted material candidates are readily accessible. Overall, I find this work to be well written and largely scientifically correct.

The proposal of OD topological dislocation states is relatively novel: an early related study was conducted in [Roy and Juricic, PRR (2021)] and at the same time the present work was posted, a very closely related proposal in normal-state insulators was also posted by Schindler et al. (and was eventually published in Nature Communications). Hence, from the perspective of novelty, this work should be treated on the same footing as [Schindler et al., Nat. Comm. (2022)], and therefore similarly rises to the level of novelty required for publication in Nat. Comm.

However I am not yet sure whether this work is sufficiently general in theoretical depth and scope to merit publication in Nature Communications, or whether it should appear in a more specialized journal. The closest point of comparison is the related work by Schindler et al., from which the present work differs in the following ways:

1. The present work considers dislocations in topological superconductors
2. Schindler et al. introduce predictive index theorems for when OD topological dislocation states appear, whereas the dislocation states in the present work depend on details beyond the bulk topology and defect Burgers vector (i.e. are “boundary obstructed”).
3. Schindler et al., provide extensive k.p, band theory, and first-principles calculations to support their calculations.
4. Schindler et al., relate their findings to magnetic flux insertion, and compare the results using the language of quantum anomalies and response.

If the authors could improve greatly on comparison points 2-4, I could potentially recommend this work for publication in Nat. Comm. As elaborated in my specific comments below, I would like to see more rigorous low-energy arguments, stronger conditions for predicting when dislocations in superconductors trap OD Majorana states (and when they do not), and calculations performed using models obtained through first-principles (DFT) calculations. If these calculations were performed, summarized in the main text, and presented in full detail in a supplemental file, I believe that this work would be a better fit for Nature Communications.

Additional Comments:

1. Throughout the text, the use of the word “etc.” is a bit too informal; I recommend that the authors not use this expression.

2. The authors write in the introduction that they are not aware of any theoretical or experimental proposal for 0D Majorana dislocations states. Does this statement only apply to dislocations in 3D SCs, or in 2D SCs as well?

3. The role of degeneracy in the nested domain wall construction should be highlighted more clearly. For example, in a 3D material, a fourfold Dirac fermion can form a first-order domain wall to give a 2D twofold Dirac fermion, and then a second-order domain wall in that 2D Dirac fermion can give a 1D chiral mode (“axion string”), as described in numerous works including (but not limited to) [Fan Zhang, Kane, Mele PRL (2013)] and [Zhong Wang and S.C. Zhang PRB (2013)]. However to get 0D states in 3D from nested domain walls, one has to start with a bulk eightfold Dirac fermion of the kind introduced in [Wieder et al., PRL (2016)]. This kind of “Dirac hierarchy” was recognized in various forms in recent years, including [Wieder et al., Science (2018)], [Li-Yang Zheng and J. Christensen, PRL (2021)], and [Linyun Yang, Yin Wang, et al., PRL (2022)], and should be discussed in relation to the earlier works.

4. The nested domain wall construction for 0D topological corner-like states should also be attributed to [Wieder et al., Nat. Comm. (2020)].

5. The use of the word “embedded” topological state in this work is a bit different than its use in the work by [Tuegel et. al., PRB (2019)] and related subsequent works. In those papers, the left-over topological states arise typically from decoupled sublattices and stacking faults, rather than dislocations with integer Burgers vectors. The distinction between the two uses of the word “embedded” should be made much more clearly in the present work as close as possible to the first use of the word “embedded.”

6. This work is quite focused on “weak” topology, but a key feature of dislocation responses is that they only depend on weak indices, and not weak topology itself. The authors should more clearly highlight this distinction in the present work, emphasizing which topological phases are weak and strong (both in the normal state, and in the topological SC state). A graphic may help here.

7. In the caption of Fig. 1, the word “and” is misspelled as “ans.”

8. By performing a rigorous low-energy calculation, the authors can explicitly solve for the wavefunctions of the exponentially-localized 0D Majorana states and determine their localization length scales in all 3 direction in terms of different system parameters. Performing this calculation explicitly would greatly help readers understand the experimental conditions necessary for more generally realizing the proposal introduced in this work.

9. In the text preceding Eq. 2, it would be helpful if the authors could more slowly and carefully explain the degeneracy of each Dirac and hinge/1D mode in order to better understand the number of states bound in each step of the domain-wall decimation.

10. In Eq. 3, it is not clear where the “mod 2” comes from. Is this due to some Z_2 topology on the Majoranas, or in the “embedded” topological state? It would be helpful if the authors could more clearly state this proof (even if it is a restatement of a result from a previous work).

11. Throughout this work, the authors discuss the “cutting plane” as an embedded subsystem that carries a topology. However for integer dislocations, the cutting plane is a gauge-dependent object: it can be deformed to any shape ending in the dislocations, and two observers may differ whether it is the region between or outside of the dislocations. The authors should discuss this issue more clearly, and then emphasize which aspects of the procedure (and result) are gauge-independent.

12. I recommend that the authors introduce a box or table element summarizing the main quantitative predictions of the present work (i.e. present clear conditions under which 0D Majorana dislocation states can be realized in the most general gapped 3D superconductor).

13. In the conclusion, the authors conjecture the existence of “defect-obstructed” topological states, in which the appearance of topological modes on a defect depends both on the topology of the bulk insulator, as well as on details of the region surrounding the defect in a potentially non-generic (“boundary-gap”-dependent) way. Though this phenomenon is related to “boundary obstructed topology,” it is important to emphasize that more generally, defect-obstructed topology appears to lie comfortably within the well-established framework for topological defect states in [Teo and Kane, PRB (2010)]. The authors should re-express the language of “defect-obstructed” topology in the language of Teo and Kane, and explain how the findings in this work fit within the framework established by Teo and Kane (or how they do not).

Reviewer #2 (Remarks to the Author):

The authors study the formation of 0D Majorana bound states (MBS) at dislocations in 3D systems characterized by weak topology, superconductivity, and surface magnetism. They use toy models which may be relevant to a variety of real materials, including high-Tc iron-based superconductors, as well as bismuth based crystals. Their discussion is supplemented by a summary of experimental results on some of these materials, suggesting they may behave as predicted by the toy models, as well as by a simulation of experimental signatures which would correlate with the formation of dislocation MBS.

While I do believe that the authors results are innovative and should be published in Nature Communications (as I will argue at the end), I will begin by saying that I disagree with the way in which the authors present the mechanism that leads to the formation of these MBS. I disagree with the association to higher-order topology, and in particular with the statement that "our mechanism exemplifies an unprecedented Majorana mechanism that is based on the second-order topology of 2D subsystems, which is in sharp contrast with earlier proposals..." In fact, as I explain below, I believe that this mechanism, while innovative and impactful, can be explained using old and well-known TI properties.

If my explanation is wrong, then I apologize and ask the authors to correct me. To show that 0D MBS must be present in the system, I think a better approach would be to (1) start from dislocations in a weak TI (WTI), and then (2) add the superconductivity and magnetism on top as perturbations. It is known since 2009 (Ref. [3]) that a loop formed by screw and edge dislocations will effectively form a 2D quantum spin-Hall (QSH) system in a WTI. For a figure showing this schematically, see Nature Communications 13, 508 (2022), Fig. 1d. The formation of MBS then arises due to the fact that the top and bottom edges of the QSH acquire a magnetic gap due to the surface magnetism, while the side edge modes, which go through the bulk, are gapped due to bulk superconductivity. When the gap changes from magnetic to superconducting, an MBS will form on the QSH edge, as can be seen, for instance, in Fig. 1 of Phys. Rev. Lett. 101, 120403 (2008). The ingredients involved in this mechanism are known for more than a decade, so I would not call it unprecedented, nor associate it to higher-order topology. The same can be said about the class DIII MBS Kramers pairs. The QSH edge mode is known to form MBS Kramers pairs due to a switch between different superconducting gaps of different sign [e.g. Fig. 1 of Phys. Rev. Lett. 117, 046804 (2016)]. I would urge the authors to include these alternate explanations in their paper.

My criticism above notwithstanding, I do believe this work should be published in Nature Communications. I believe that showing how a well-known but overlooked mechanism may lead to MBS in a wide range of realistic systems is a powerful result. Of particular relevance is the fact that the switch from magnetic to superconducting gap that transforms the QSH modes into MBS happens due to natural properties of the crystal: intrinsic superconductivity and surface magnetism. This avoids the complications relating to building heterostructures, inducing strong and/or sharp enough

superconducting proximity gaps, etc. Beyond this, the topological phases that may form at crystal defects are actively investigated in recent years, making this result timely. While it is impossible to know in advance, I do believe this work will have a significant impact, leading to multiple follow-up theoretical as well as experimental works.

Reviewer #3 (Remarks to the Author):

In the manuscript, Hu and Zhang studied the trapped Majorana quasiparticles at lattice dislocations of topological iron-based superconductors. The key idea is that when a three-dimensional normal state has a weak band topology together with the bulk superconductivity and surface ferromagnetism, a two-dimensional subsystem spanned by screw or edge dislocations can be considered as an embedded second order topological superconductor hosting Majorana corner modes. The theory is further extended to the system without surface magnetism. The authors also carefully discussed the candidate materials and experimental methods for detecting the Majorana states using STM experiment.

Overall, the manuscript is well-written, and especially, the mechanism generating Majorana modes at lattice dislocation is clearly presented. The implication or relevance of the theoretical idea to real iron-based materials is carefully described by using the model Hamiltonian related to iron-based superconductor.

However, the relation between the weak band topology and dislocation Majorana mode is already known. For example, in [T.L Hughes et al., "Majorana zero modes in dislocations of Sr₂RuO₄", PRB 90, 235123(2014)], [S. B. Chung et al., "Dislocation Majorana zero modes in perovskite oxide 2DEG", Scientific Reports 6, 25184 (2016)], dislocation Majorana zero modes are carefully studied and its relevance to real materials is also thoroughly discussed. Although the present manuscript contains new development to a certain level compared to these preceding works, the progress looks like incremental, not at the level of fundamental development. Also, the required conditions to observe the dislocation Majorana modes proposed in this manuscript are quite complicated. Simultaneous presence of weak bulk topology, controlled dislocation, uniform surface ferromagnetism is difficult to be achieved in real materials in a manner that the proposed theoretical idea can actually be observed even in iron-based materials. In this respect, I do not think that the novelty of this paper is at the level of Nature Communications. Let me give a couple of additional questions/comments below.

(1) In Figure 1 (b), it is better to mark the axes for real space coordinates. In Figure 1 (c), is $t=t(x)$ correct? $t=t(y)$ looks suitable.

(2) On page 3, in the paragraph starting with “To explore the fate of chiral Majorana hinge modes during the gluing process,...”. The third sentence should be “Distribution of δ_M and δ_{SC} are shown in Fig.1(c).

(3) Related to the sentence “However, the interlayer mass term for the Majorana fermions will obtain a phase factor $e^{i\pi b\nu}$, following the side Dirac surface states”, I suggest to add a couple of sentences describing the origin of the phase factor.

(4) The meaning of the term “dislocation dipole” should be more clearly presented.

(5) The description of the inflation of a Majorana mode from 0D to 1D seems to be valid only when the circular radius of the cylindrical geometry is small. I do not understand how meaningful this description is in bulk materials.

(6) Related to the Figure 3 and related description, it is stated that PFS and ET2 phases can coexist. However, when the bulk is gapless, I am wondering how stable the Majorana modes are and how their spatial localization is affected by. Also, what is the qualitative difference between Figure 3 (c) and (d)? It is unclear what kind physics is intended to be delivered by comparing these two panels.

(7) The discussion about class DIII ET2 is rather brief. How the bulk s_{\pm} pairing can induce a pairing mass domain for Dirac fermions?

(8) As for the CdGM states inside the SC gap, it is stated that ET2 dislocation core carries fewer or even no subgap states besides the dMBS. What is the physical reason for it?

(9) I think that more discussion is necessary related to the spatial anisotropy of dMBS wave function. How is the anisotropic shape related to the in-plane mirror symmetry? I think the same symmetry breaking effect also exists in vortex Majorana wave function that is usually circular symmetric. I think the issue is the length scale of the spatial localization of the wave function, not related to whether the Majorana is bound to a dislocation or a magnetic vortex.

(10) How the non-Abelian nature of the dislocation Majorana zero mode can be experimentally detected?

(11) It is stated that ET2 phase is “extrinsic”. Why is it impossible to characterize dMBS by 2D bulk topological invariant of the subsystem? Any fundamental reason for it?

Dear Reviewers,

Thank you so much for your time and expertise to review our manuscript, and for offering invaluable suggestions that have played a pivotal role in enhancing its quality. Before our point-to-point response to your comments, please allow us to provide a concise overview of the revisions we implemented.

✓ Major reversions we did:

- (1) We have derived the low-energy theory for dislocation Majorana modes. This effort not only provides a concrete understanding of the realization of the 0D dislocation Majorana zero mode (MZM), but also leads us to an analytical formula for the localization length of the MZM.
- (2) We have conducted numerical simulations to investigate the localization length of MZMs, which are in great agreement with our analytical predictions.
- (3) We have studied a realistic model for $\text{FeTe}_{0.5}\text{Se}_{0.5}$ based on density-functional-theory results, based on which we have numerically obtained a topological phase diagram for ET_2 . It demonstrates a remarkable agreement with the outcomes of the toy model presented in the main text.
- (4) We have investigated the fate of dislocation MZMs in the presence of strong spatial magnetization fluctuations, and confirmed their remarkable resilience.
- (5) Three new supplementary notes are now included to summarize the above discussions. We have also performed substantial revisions to the main text (which are highlighted in blue). A detailed summary of major changes can be found at the very end of this reply letter.

With these significant reversions, we believe all the comments from all reviewers have been fully addressed. We believe the revised manuscript has met the high standards in Nature Communications and is suitable for immediate publication.

Finally, we sincerely apologize for the unexpected delay in submitting the response letter. The past year has been challenging for us, especially since the first author has been experiencing multiple job applications and relocations, as well as important updates to his personal life. This has led to extensive travel among the U.S., China, and Europe over the past year, which partially accounts for this unexpected delay. Besides, we have spent a significant amount of time and effort to revise and improve the manuscript, which now includes a new 14-page supplementary material. We sincerely hope that this delay will not adversely impact your decision of our work in Nature Communications.

Reviewer #1

Remarks to the Author

In this work, the authors demonstrate that under certain conditions, a lattice dislocation in a topological superconductor will bind 0D Majorana states. The authors support their findings with theoretical arguments, tight-binding calculations, and material analysis. The material realization discussion in particular is quite thoughtful, and the highlighted material candidates are readily accessible. Overall, I find this work to be well written and largely scientifically correct.

The proposal of 0D topological dislocation states is relatively novel: an early related study was conducted in [Roy and Juricic, PRR (2021)] and at the same time the present work was posted, a very closely related proposal in normal-state insulators was also posted by Schindler et al. (and was eventually published in Nature Communications). Hence, from the perspective of novelty, this work should be treated on the same footing as [Schindler et al., Nat. Comm. (2022)], and therefore similarly rises to the level of novelty required for publication in Nat. Comm.

However I am not yet sure whether this work is sufficiently general in theoretical depth and scope to merit publication in Nature Communications, or whether it should appear in a more specialized journal. The closest point of comparison is the related work by Schindler et al., from which the present work differs in the following ways:

Q1. The present work considers dislocations in topological superconductors

Q2. Schindler et al. introduce predictive index theorems for when 0D topological dislocation states appear, whereas the dislocation states in the present work depend on details beyond the bulk topology and defect Burgers vector (i.e. are “boundary obstructed”).

Q3. Schindler et al., provide extensive k.p, band theory, and first-principles calculations to support their calculations.

Q4. Schindler et al., relate their findings to magnetic flux insertion, and compare the results using the language of quantum anomalies and response.

If the authors could improve greatly on comparison points 2-4, I could potentially recommend this work for publication in Nat. Comm. As elaborated in my specific comments below, I would like to see more rigorous low-energy arguments, stronger conditions for predicting when dislocations in superconductors trap 0D Majorana states (and when they do not), and calculations performed using models obtained through first-principles (DFT) calculations. If these calculations were performed, summarized in the main text, and presented in full detail in a supplemental file, I believe that this work would be a better fit for Nature Communications.

Reply: We greatly appreciate Reviewer #1 for carefully reading our manuscript and his/her positive assessment of our work, as well as the inspiring comments. The reviewer has mentioned the work by Schindler et al. that was earlier published in Nat Commun 13, 5791 (2022) (i.e., cited as Ref. [62] in the main text), which has served as a useful guidance for us to improve the quality of our work. Here, we would like to first clarify a few key differences between Ref. [62] and our work, and then move to a brief summary of the revisions we have made to address the reviewer’s questions. Finally, we will provide a point-to-point response to the additional comments by Reviewer #1.

First of all, our work is concerned with emergent defect-enabled topological physics in a **topologically trivial** bulk superconductor. While band topology has been proved ubiquitous in electronic systems, finding a convincing candidate of topological superconductor (TSC) has faced multiple well-known challenges. This is the key practical reason for us to focus on “trivial” superconductors, but not their topological cousins, in this work.

The absence of bulk-state topology has set our work apart from most (if not all) of the previous works on the dislocation-trapped bound states, including Ref. [62]. This is why in our work, a bulk index theorem like the one discovered in Ref. [62] is not expected. Instead, we have an alternative criterion that depends on the interplay of the Burgers vector and the 2D BdG Chern number of surface states. It is crucial to notice that the bulk superconducting state with the *s*-wave spin-singlet pairing itself **lacks any nontrivial weak \mathbb{Z}_2 index at the BdG level**. Directly applying the theory established in Ref. [62], the anticipation of 0D dislocation Majorana zero modes (MZM) is not warranted.

This is also consistent with the fact that the “embedded topology” encoded in the dislocation cutting plane is “extrinsic”, but not “intrinsic” as the ones revealed in Ref. [62]. In this regard, the dislocation physics in our work is also less relevant to a quantum anomaly, given the triviality of bulk states. Nonetheless, in the main text, we have discussed the physical consequence of superconducting vortices in our system, which is perhaps the best analogy to a magnetic flux tube in a non-superconducting topological system.

To clarify these differences with prior works, we have added a new table as the reviewer suggested in the main text to highlight the topological properties and boundary modes for both normal-state and superconducting state of $\text{FeTe}_{0.5}\text{Se}_{0.5}$ (see also reply to the additional **Comment 6** below). Following the reviewer’s suggestion, we have also developed a low-energy effective theory for the condition to realize the 0D dislocation Majorana zero modes (see the response to the additional **Comment 8** below). It further leads to analytical expressions for the localization length of dislocation MZMs. Together with the necessary revisions we have made in the main text, our above efforts should have fully addressed the reviewer’s **Comments Q1, Q2, and Q4** above.

Regarding your **Comment Q3** above, the reviewer has raised an excellent point regarding a DFT-based analysis for $\text{FeTe}_{0.5}\text{Se}_{0.5}$ (FTS). As elaborated in the new supplementary materials, we have carefully analyzed a first-principles-based $\mathbf{k} \cdot \mathbf{p}$ model for FTS to study the topological phase diagram for dislocation MZM. Remarkably, the new calculation results based on the DFT-based model are **fully consistent** with our prior results based on the effective 4-band model. A brief summary is provided below:

✓ **The normal-state band structure.**

In the Supplementary Note 1.1, we first exploited the DFT-fitted effective model to quantitatively study the surface magnetic gap for FTS. By building a slab geometry along (001) direction, the spectrum along k_x axis with $M_z = 0$ is shown in Fig. R1 (a), where the surface Dirac cone appears in a narrow energy windows $[-6.394, 6.831]$ meV. At the Fermi level, note that other bulk bands also contribute to the Fermi surface (see Gang Xu et al., *Phys. Rev. Lett.* **117**, 047001 (2016) for details). As a comparison, the spectrum along the k_x axis in Fig. R1 (b) shows the gapped Dirac cones by surface magnetism with $M_z = 5$ meV. We also show the localized wave function distribution of the Dirac surface state in Fig. R1 (c), showing that the localization length is about 30 layers (~ 18 nm). The localization length is much larger than the typical scale in

Bi_2Se_3 and Bi_2Te_3 ($\sim 3 \text{ nm}$) due to the small SOC strength.

Figure R1: **Band structures and surface topological phase diagram.** Based on the DFT-based model for $\text{FeTe}_{0.5}\text{Se}_{0.5}$, we first reproduce the topological surface states within a topological gap $[-6.394, 6.831 \text{ meV}]$, depicted in (a). Upon activating surface magnetization ($M_z = 5 \text{ meV}$), a distinct magnetic gap emerges for the Dirac cone in (b). And (c) illustrates the localized wave function distribution of the Dirac surface states, the localization length in the c -axis is about 30 layers ($\sim 18 \text{ nm}$). As $\text{FeTe}_{0.5}\text{Se}_{0.5}$ transitions into the superconducting phase, a proximity-induced superconducting gap ($\sim 1.82 \text{ meV}$) emerges for the Dirac surface states, as illustrated in (d). The interplay between surface magnetism and superconductivity is evident in the spectrum depicted in (e). Additionally, the topological phase diagram, computed numerically with respect to chemical potential μ and M_z , is presented in (f). At low M_z , it exhibits a trivial phase and transforms into an embedded second-order topological superconductor housing 0D dislocation Majorana zero modes upon the increment of M_z . The color representation in (f) depicts the logarithm of the gap of the Dirac surface states at the Γ point. It is in excellent agreement with the results based on the effective model discussed in the main text (see Fig. 3 (b)).

✓ **The superconducting band structure.**

In the Supplementary Note 1.2, we discuss the BdG Hamiltonian for FTS. Note that only s-wave spin-singlet pairing is considered here, so that the bulk ground state is *not topological*. The band structure without surface magnetism is shown in Fig. R1 (d), where the proximity-induced superconducting gap is about 1.82 meV , being quantitatively consistent with recent experimental observations [Peng Zhang et al., science 360, 6385 (2018)]. Furthermore, the time-reversal symmetry protected Kramers degeneracy

at the Γ point is broken after turning on the surface magnetism that competes with the superconductivity, as shown in Fig. R1 (e).

✓ **Topological surface phase diagram for 0D dislocation Majorana modes.**

We hope to first comment on the *infeasibility* of performing a full 3D simulation for the dislocation Majorana modes with the DFT-based model. Given the meV-level topological gap here, the Majorana localization length is estimated to be around hundreds of unit cells. To avoid the finite-size effect here, a proper full 3D geometry could consist of millions of unit cells, leading to a BdG Hamiltonian matrix with a dimensionality of more than 10 million. Unfortunately, this is far beyond the capacity of our existing computing resources. This is a common challenge in the context of superconductivity but maybe not to topological electronic systems, which simply because the superconducting gap is usually several orders of magnitude smaller than an electronic band gap.

In spite of this challenge, the surface topological phase diagram manifests as an equivalent yet much more efficient way to identify the existence of these dislocation Majorana modes, based on the topological theory we established in this work. The topological conditions have been clearly demonstrated via the low-energy effective theory. Therefore, in the Supplementary Note 1.3, we numerically map out the surface topological phase diagram as a function of chemical potential μ and magnetism strength M_z , and the results are shown in Fig. R1 (f). Quite remarkably, **this new DFT-based phase diagram exactly resembles our prior result shown in Fig. 3 (b) of the main text.** This provides a solid proof that our theory for dislocation Majorana modes is directly applicable to FTS samples in the real world.

Hence, we are confident that these revisions have adequately addressed the major comments raised by the reviewer.

A Point-to-Point Response to Additional Comments

Comment 1: Throughout the text, the use of the word “etc.” is a bit too informal; I recommend that the authors not use this expression.

Reply: We have followed the suggestions to remove the “etc” in the revised manuscript.

Comment 2: The authors write in the introduction that they are not aware of any theoretical or experimental proposal for 0D Majorana dislocations states. Does this statement only apply to dislocations in 3D SCs, or in 2D SCs as well?

Reply: We thank the reviewer for bringing out this question. In fact, there exist a handful of theoretical proposals (i.e., Ref. [20 - 22]) which focus on 2D and 3D p-wave topological superconductors with weak \mathbb{Z}_2 index. In some sense, they can be viewed as a superconducting analogue of the original dislocation theory for weak topological insulators. However, given the scarcity of p-wave superconductors, there has been no trace of dislocation Majorana modes

in experiments. Therefore, we stated in the previous introduction that “*we are not aware of any realistic superconducting system that has been theoretically proposed or experimentally supported to feature Majorana dislocation-bound states*”, as we found the previous theoretical works not practically feasible. This statement is indeed misleading, and we have performed a revision in the introduction to clarify the research status of dislocation Majorana physics.

Comment 3: The role of degeneracy in the nested domain wall construction should be highlighted more clearly. For example, in a 3D material, a fourfold Dirac fermion can form a first-order domain wall to give a 2D twofold Dirac fermion, and then a second-order domain wall in that 2D Dirac fermion can give a 1D chiral mode (“axion string”), as described in numerous works including (but not limited to) [Fan Zhang, Kane, Mele PRL (2013)] and [Zhong Wang and S.C. Zhang PRB (2013)]. However to get 0D states in 3D from nested domain walls, one has to start with a bulk eightfold Dirac fermion of the kind introduced in [Wieder et al., PRL (2016)]. This kind of “Dirac hierarchy” was recognized in various forms in recent years, including [Wieder et al., Science (2018)], [Li-Yang Zheng and J. Christensen, PRL (2021)], and [Linyun Yang, Yin Wang, et al., PRL (2022)], and should be discussed in relation to the earlier works.

Reply: We thank the reviewer for pointing out these interesting references. We have included them in the revised manuscript.

Comment 4: The nested domain wall construction for 0D topological corner-like states should also be attributed to [Wieder et al., Nat. Comm. (2020)].

Reply: We thank the reviewer for pointing out this interesting reference. We have included it in the revised manuscript.

Comment 5: The use of the word “embedded” topological state in this work is a bit different than its use in the work by [Tuegel et. al., PRB (2019)] and related subsequent works. In those papers, the left-over topological states arise typically from decoupled sublattices and stacking faults, rather than dislocations with integer Burgers vectors. The distinction between the two uses of the word “embedded” should be made much more clearly in the present work as close as possible to the first use of the word “embedded.”

Reply: We agree with the reviewer about the difference between our work and this PRB paper. In the work by Tuegel et al., the embedded topology is intrinsic, in the sense that the 2D cutting plane itself carries a bulk topological index. While in our work, to borrow the term of higher-order topological physics, the embedded topology is “extrinsic” and a probably more accurate description would be the domain-wall topology. We have clarified this issue in the revised manuscript.

Comment 6: This work is quite focused on “weak” topology, but a key feature of dislocation responses is that they only depend on weak indices, and not weak topology itself. The

authors should more clearly highlight this distinction in the present work, emphasizing which topological phases are weak and strong (both in the normal state, and in the topological SC state). A graphic may help here.

Reply: We thank the reviewer for this suggestion. We agree with the reviewer that the weak indices are more essential and accurate than weak topology in our theory. We have followed the reviewer’s suggestion to add a new table in the main text to clarify the topological properties at each stage in our theory.

Comment 7: In the caption of Fig. 1, the word “and” is misspelled as “ans.”

Reply: We thank the reviewer for pointing out this typo, which we have fixed in the revised manuscript.

Comment 8: By performing a rigorous low-energy calculation, the authors can explicitly solve for the wavefunctions of the exponentially-localized 0D Majorana states and determine their localization length scales in all 3 direction in terms of different system parameters. Performing this calculation explicitly would greatly help readers understand the experimental conditions necessary for more generally realizing the proposal introduced in this work.

Reply: We thank the reviewer for this great suggestion! In the supplementary material, we have developed such a low-energy theory for the dislocation Majorana bound states and extracted the localization length analytically. Further numerical calculations of the localization length agree well with our analytical results. Below, we provide a summary of both results:

(1) Analytical approach for localization length (see Supplementary Note 2.1).

For the topological surface state, we obtain the localized mode near $z = 0$ boundary between sample ($z \geq 0$) and vacuum ($z < 0$)

$$\psi^\uparrow(z) = 2ce^{-Az} \sin(Bz) |\phi_-\rangle \otimes |\uparrow\rangle, \quad (\text{R1a})$$

$$\psi^\downarrow(z) = 2ce^{-Az} \sin(Bz) |\phi_-\rangle \otimes |\downarrow\rangle. \quad (\text{R1b})$$

where $|c| = \sqrt{\frac{A(A^2+B^2)}{2A^2+B^2}}$ is the normalization factor with $A = \frac{v}{2m_2}$ and $B = \sqrt{2m_{0z}m_2 - v^2}$. The spinor part provides the basis for Dirac surface states, $\{|\phi_-\rangle \otimes |\uparrow\rangle, |\phi_-\rangle \otimes |\downarrow\rangle\}$, where $\sigma_x|\phi_\pm\rangle = \pm|\phi_\pm\rangle$ leads to $|\phi_+\rangle = \frac{1}{\sqrt{2}}(1, 1)^T$ and $|\phi_-\rangle = \frac{1}{\sqrt{2}}(1, -1)^T$. Thus, the surface state Hamiltonian up to the linear k order is

$$\mathcal{H}_{surf} = v(k_x s_y - k_y s_x). \quad (\text{R2})$$

Please note that \mathbf{s} are Pauli matrices for the spin degree of freedom. The localization length for the surface state is given by $\xi_{surf} = 1/A = 2m_2/v$, which indicates that a small spin-orbit coupling (v) will correspond to a large localization length.

On the sample surface, the ferromagnetism coexists with bulk superconductivity, thus the BdG Hamiltonian for top surface of the sample can be described as

$$\mathcal{H}_{BdG} = v(k_y s_y \gamma_z - k_x s_x \gamma_0) - \mu s_0 \gamma_z + M_z s_z \gamma_z + \Delta_0 s_y \gamma_y, \quad (\text{R3})$$

Figure R2: **Schematic for the analytical solution of surface states, chiral Majorana zero modes (MZMs), and dislocation MZMs.** (a) illustrates the localized Dirac surface states of a topological insulator. (b) showcases the interplay between surface magnetism and bulk superconductivity, resulting in chiral Majorana zero modes (MZMs) represented by blue lines along the hinges of the top and bottom surfaces. (c) displays the cutting plane formed by a pair of dislocation lines. The corresponding top view is also presented on the top surface in (d), where each side features one chiral MZM mode (the triangle indicates the propagating direction). The final step involves gluing these two sections together to restore the lattice, leading to the hybridization between the two chiral MZMs and the emergence of 0D dislocation MZMs.

where γ are Pauli matrices for the particle-hole degree of freedom. Without loss of generality, we consider both $M_z > 0$ and $\Delta_0 > 0$. The gap closing of the quasi-particle happens at $\mu^2 + \Delta_0^2 = M_z^2$ for the Γ point, and the topological gap is given by $\Delta_{topo} = M_z - \sqrt{\mu^2 + \Delta_0^2}$. For the topological phase of the sample surface, the chiral MZM solution for $\mathcal{H}_{BdG}(k_x = 0, -i\partial_y)$ is

$$\psi(y) = ae^{(-M_z + \Delta_0)y/v} |\phi_+\rangle, \quad (\text{R4})$$

where a is the normalization factor and the spinor part is $|\phi_+\rangle = \frac{1}{2}(1, -i, -i, 1)^T$. It is eigen-state of chiral symmetry $s_y\gamma_z|\phi_+\rangle = -|\phi_+\rangle$, and eigen-state of particle-hole symmetry $s_0\gamma_xK|\phi_+\rangle = i|\phi_+\rangle$. It is also eigen-state of the k_x term ($k_x s_x \gamma_0$ in \mathcal{H}_{BdG}), which gives rise to the dispersion of 1D chiral MZM along the 1D boundary of the sample surface, namely, $\langle\phi_+|k_x s_y \gamma_z|\phi_+\rangle = k_x$.

We next solve the dislocation MZM by inserting a pair of dislocation lines to the system. As we discussed in the main text, the dislocation line is aligned to the z -direction. There are two main steps.

- Cut the sample into two parts by the cutting plane expanded by this pair of dislocation lines, as illustrated in Fig. R2 (c) [see the dashed rectangle]. The 1D chiral MZM is also divided into two parts, and near the touching edges, those two 1D chiral MZMs propagate along opposite direction due to the C_{2z} symmetry [see Fig. R2 (d)].
- Glue these two 1D chiral MZMs by restoring the lattice, as illustrated in Fig. R2 (e).

After these two steps, we can build an effective two-by-two Hamiltonian consisting of two oppositely propagating 1D chiral MZMs, which are

$$\psi_R(y) \approx e^{(-M_z + \Delta_0)y/v} (1, -i, -i, 1)^T, \quad (\text{R5a})$$

$$\psi_L(y) \approx e^{(M_z - \Delta_0)y/v} (1, i, i, 1)^T. \quad (\text{R5b})$$

This gives rise to

$$\mathcal{H}_{dis} = vk_x \tau_z + \text{Im}[t_c] \tau_y + \text{Re}[t_c] \tau_x, \quad (\text{R6})$$

where the inter-edge coupling t_c is due to the direct hopping ($vk_y s_x \gamma_0 \rightarrow -iv \partial_y s_x \gamma_0$),

$$\begin{aligned} t_c &\approx \int_{-\Delta_y}^{\Delta_y} dy \langle \phi_R | s_x \gamma_0 | \phi_L \rangle \times (e^{(-M_z + \Delta_0)y/v} [-iv \partial_y] e^{(M_z - \Delta_0)y/v}) \\ &\propto -iv \frac{M_z - \Delta_0}{v} = -i(M_z - \Delta_0). \end{aligned} \quad (\text{R7})$$

Please also note that t_c is just a constant if there are no dislocation pairs, however, its sign depends on the position for the dislocation case [c.f., Ying Ran, et al., Nature Physics 5, 298–303 (2009)]. For example, these two dislocation lines are separated in real space and locate at $(N_{cx}, N_{cy}, z = 1) \rightarrow (N_{cx}, N_{cy}, z = N_z)$ [line 1] and $(N_{cx} + \Delta_x, N_{cy}, z = 1) \rightarrow (N_{cx} + \Delta_x, N_{cy}, z = N_z)$ [line 2]. The in-plane distance between these two dislocation lines is given by Δ_x . Then, we have

$$t_c = \begin{cases} x \leq N_{cx} \text{ or } x \geq N_{cx} + \Delta_x, & \text{direct in-plane hopping: } -i(M_z - \Delta_0), \\ N_{cx} \geq x \leq N_{cx} + \Delta_x, & \text{inter-layer hopping: } -i(M_z - \Delta_0) \times (-1). \end{cases} \quad (\text{R8})$$

The hopping for spin-orbit coupling acquires a π phase shift on circling the dislocation, which explains the minus above. Therefore, t_c is the mass term for the two 1D chiral MZMs, and changes sign. It leads to the 1D domain wall along the line connecting these two dislocation lines. A 0D MZM, called dislocation MZM in the main text, can naturally appear, whose in-plane localization length is given by

$$\xi_{MZM} \approx \frac{v}{M_z - \Delta_0}. \quad (\text{R9})$$

This implies that, ξ_{MZM} increases when we increase v or Δ_0 , note that $M_z > \Delta_0$ is required. Beside, it increases when we decrease M_z . Turning on finite chemical potential μ only changes the topological gap, so that we argue

$$\xi_{MZM} \approx \frac{v}{M_z - \sqrt{\Delta_0^2 + \mu^2}}. \quad (\text{R10})$$

Figure R3: **The analytical result for the localization length (ξ_{MZM}) of dislocation Majorana zero modes (MZMs).** We illustrate the relationship between ξ_{MZM} and three key parameters: the Dirac surface state's velocity v in (a), the proximity-induced superconducting gap Δ_0 in (b), and the surface magnetization strength M_z in (c).

A plot of Eq. (R10) is shown in Fig. R3 for illustrations. Please note that Eq. (R10) can only qualitatively capture the in-plane localization length, which is also the one that is experimentally measurable using scanning tunneling microscopy.

(2) **Numerical approach for localization length.** We note that there exist two physically equivalent measures to numerically quantify the localization lengths for the dislocation MZMs in our system,

- Extracting the localization length via finite-size gap (see Supplementary Note 2.2). This method is illustrated in Fig. R4 (a) and (b).
- Fitting the localization length via the wave function (see Supplementary Note 2.3). This method is illustrated in Fig. R4 (c) and (d).

Below, we focus on the **approach I**, to use the numerical simulation for extracting the finite-size gap Δ_{MZM} of a pair of dislocation MZMs based on the exact diagonalization of the 3D lattice Hamiltonian. As shown in Fig. R4 (a), we only consider top surface magnetism, which leads to two dislocation MZMs localized at the top end of each dislocation line. The two gray cones in Fig. R4 (a) denote the dislocation MZMs. The ignorance of bottom surface magnetism and the consequential MZMs can significantly reduce the required lattice size along the z direction in our large-scale simulations. Therefore, the finite size gap Δ_{MZM} is mainly caused by the hybridization between the two top dislocation MZMs, providing a relatively accurate in-plane localization length of MZMs. Varying the in-plane distance between these two dislocation lines, labeled by Δ_x in Fig. R4 (a), we can obtain the curve for Δ_{MZM} as a function of Δ_x , as shown in Fig. R4 (b) for an example.

In the numerical calculation, we set $N_y = 2N_x = 2\Delta_x \in [28, 56]$ and $N_z = 16$. Even though the calculation on the lattice size with $N_y = 2N_x = 56$ and $N_z = 16$ almost reaches our numerical computing ability, this method shows enough data to fit ξ_{MZM} .

Figure R4: **Illustration of two approaches for estimating the localization length (ξ_{MZM}) of dislocation Majorana zero modes (MZMs).** (a) represents the approach I (finite-size effect) by numerical calculating the hybridization gap Δ_{MZM} between two dislocation MZMs. For example, the corresponding calculation is presented in (b), showing Δ_{MZM} as a function of the in-plane distance between them, denoted as Δ_x . The numerical results are represented by blue dots, and the black line indicates the fitting result. From this fit, we extract $\xi_{MZM} \sim 5.62057$ in the unit of the in-plane lattice constant. Furthermore, (c) demonstrates approach II by extracting ξ_{MZM} along various directions from the wave function distribution of a single dislocation MZM, as depicted in (d). Likewise, the numerical results are represented by blue dots, and the black line indicates the fitting result.

An example is shown in Fig. R4 (b), where the blue dots represent numerical results, and can be fitted by an exponential function,

$$\Delta_{MZM} \approx ae^{-\Delta_x/\xi_{MZM}}. \quad (\text{R11})$$

For the result in Fig. R4 (b), the black line is for the fitting results and shows the in-plane localization length of MZMs as $\xi_{MZM} = 5.62057$ in the unit of in-plane lattice constant.

Based on this, we can further calculate the dependence of ξ_{MZM} on tuning parameters, including spin-orbit coupling v , superconducting gap Δ_0 , and surface magnetism M_z . Please notice that M_z and Δ_0 here are directly added in the 3D bulk lattice Hamiltonian, which can be slightly different from the corresponding projected values for Dirac surface states, as used in the analytical approach for localization length. The numerical results are shown in Fig. R5, which shows a great agreement with analytical results in Eq. (R10) [also see Fig. R3]. Increasing both v and Δ_0 gives rise to the increase of ξ_{MZM} , but increasing M_z leads to the decrease of ξ_{MZM} .

Figure R5: **The numerical results for the localization length (ξ_{MZM}) of dislocation Majorana zero modes (MZMs).** Similar to the analytical results of ξ_{MZM} shown in Fig. R3 above, here we use the approach I to numerically calculate ξ_{MZM} as a function of different tuning parameters, the spin-orbit coupling v in (a), the bulk superconducting gap Δ_0 in (b), and the surface magnetization strength M_z in (c).

Comment 9: In the text preceding Eq. 2, it would be helpful if the authors could more slowly and carefully explain the degeneracy of each Dirac and hinge/1D mode in order to better understand the number of states bound in each step of the domain-wall decimation.

Reply: We thank the reviewer for this suggestion. We have revised our main text accordingly.

Comment 10: In Eq. 3, it is not clear where the “mod 2” comes from. Is this due to some \mathbb{Z}_2 topology on the Majoranas, or in the “embedded” topological state? It would be helpful if the authors could more clearly state this proof (even if it is a restatement of a result from a previous work).

Reply: An explanation of this “mod 2” question can be found in our response to the reviewer’s **Comment 8** above. In particular, the hopping between two chiral Majorana mode propagating along different direction (see Fig. R2 (d)) in Eq. (R8) changes sign due to the presence of screw dislocation. This microscopic hopping mechanism resembles that was originally discussed in Ref. [Ying Ran et al., *Nature Physics* 5, 298–303 (2009)]. This highlights the necessity for a nonzero weak \mathbb{Z}_2 index in the bulk normal-state Hamiltonian to facilitate the realization of 0D dislocation Majorana zero modes even in a trivial bulk superconductor.

Comment 11: Throughout this work, the authors discuss the “cutting plane” as an embedded subsystem that carries a topology. However for integer dislocations, the cutting plane is a gauge-dependent object: it can be deformed to any shape ending in the dislocations, and two observers may differ whether it is the region between or outside of the dislocations. The authors should discuss this issue more clearly, and then emphasize which aspects of the procedure (and result) are gauge-independent.

Reply: This is an interesting point! Indeed, the cutting plane is gauge-dependent as the reviewer pointed out. Notably, the cutting plane in our system carries an extrinsic 2nd-order topology but not an intrinsic one. As a result, the presence of dislocation Majorana modes in our theory is determined by the domain wall configuration around the dislocation, rather than the topological index carried by the cutting plane (which is trivial). Therefore, the dislocation Majorana modes in our theory are indeed gauge-independent. We have clarified this point in the revised manuscript.

Comment 12: I recommend that the authors introduce a box or table element summarizing the main quantitative predictions of the present work (i.e. present clear conditions under which 0D Majorana dislocation states can be realized in the most general gapped 3D superconductor).

Reply: We thank the reviewer for this suggestion. The condition for 0D dislocation Majorana zero mode and the main results are actually summarized in Fig. 1, supported by numerical calculations. We have followed your **Comment 6** above to add a table for clarifying the topology of normal-state, superconducting bulk-state and superconducting surface-state. We hope this update will greatly improve the readability of our manuscript.

Comment 13: In the conclusion, the authors conjecture the existence of “defect-obstructed” topological states, in which the appearance of topological modes on a defect depends both on the topology of the bulk insulator, as well as on details of the region surrounding the defect in a potentially non-generic (“boundary-gap”-dependent) way. Though this phenomenon is related to “boundary obstructed topology”, it is important to emphasize that more generally, defect-obstructed topology appears to lie comfortably within the well-established framework for topological defect states in [Teo and Kane, PRB (2010)]. The authors should re-express the language of “defect-obstructed” topology in the language of Teo and Kane, and explain how the findings in this work fit within the framework established by Teo and Kane (or how they do not).

Reply: We appreciate the reviewer for this suggestion. We agree with the reviewer that the discussion of “defect-obstructed” topological states are completely beyond the scope of this work. Thus, we have removed the relevant statement in the main text to avoid further confusions to the readers. We will leave a further investigation of this matter to future works.

Remarks to the Author

The authors study the formation of 0D Majorana bound states (MBS) at dislocations in 3D systems characterized by weak topology, superconductivity, and surface magnetism. They use toy models which may be relevant to a variety of real materials, including high-Tc iron-based superconductors, as well as bismuth based crystals. Their discussion is supplemented by a summary of experimental results on some of these materials, suggesting they may behave as predicted by the toy models, as well as by a simulation of experimental signatures which would correlate with the formation of dislocation MBS.

While I do believe that the authors results are innovative and should be published in Nature Communications (as I will argue at the end), I will begin by saying that I disagree with the way in which the authors present the mechanism that leads to the formation of these MBS. I disagree with the association to higher-order topology, and in particular with the statement that “our mechanism exemplifies an unprecedented Majorana mechanism that is based on the second-order topology of 2D subsystems, which is in sharp contrast with earlier proposals..” In fact, as I explain below, I believe that this mechanism, while innovative and impactful, can be explained using old and well-known TI properties.

If my explanation is wrong, then I apologize and ask the authors to correct me. To show that 0D MBS must be present in the system, I think a better approach would be to (1) start from dislocations in a weak TI (WTI), and then (2) add the superconductivity and magnetism on top as perturbations. It is known since 2009 (Ref. [3]) that a loop formed by screw and edge dislocations will effectively form a 2D quantum spin-Hall (QSH) system in a WTI. For a figure showing this schematically, see Nature Communications 13, 508 (2022), Fig. 1d. The formation of MBS then arises due to the fact that the top and bottom edges of the QSH acquire a magnetic gap due to the surface magnetism, while the side edge modes, which go through the bulk, are gapped due to bulk superconductivity. When the gap changes from magnetic to superconducting, an MBS will form on the QSH edge, as can be seen, for instance, in Fig. 1 of Phys. Rev. Lett. 101, 120403 (2008). The ingredients involved in this mechanism are known for more than a decade, so I would not call it unprecedented, nor associate it to higher-order topology. The same can be said about the class DIII MBS Kramers pairs. The QSH edge mode is known to form MBS Kramers pairs due to a switch between different superconducting gaps of different sign [e.g. Fig. 1 of Phys. Rev. Lett. 117, 046804 (2016)]. I would urge the authors to include these alternate explanations in their paper.

My criticism above notwithstanding, I do believe this work should be published in Nature Communications. I believe that showing how a well-known but overlooked mechanism may lead to MBS in a wide range of realistic systems is a powerful result. Of particular relevance is the fact that the switch from magnetic to superconducting gap that transforms the QSH modes into MBS happens due to natural properties of the crystal: intrinsic superconductivity and surface magnetism. This avoids the complications relating to building heterostructures, inducing strong and/or sharp enough superconducting proximity gaps, etc. Beyond this, the topological phases that may form at crystal defects are actively investigated in recent years, making this result timely. While it is impossible to know in advance, I do believe this work will have a significant impact, leading to multiple follow-up theoretical as well as experimental works.

Reply: We greatly appreciate the reviewer for his/her recommendation of our work! The reviewer has raised an excellent point about understanding the origin of dislocation Majorana bound states (dMBSs), which, in fact, was exactly our original intuition about this system. Indeed, this QSH analog was of great help during the early stage of our project, which made us confident about the existence of dMBS even before any concrete calculations. In fact, if the target system is a weak TI, we agree with the reviewer that the QSH picture will faithfully capture the core physics here.

However, our target system $\text{FeTe}_{0.5}\text{Se}_{0.5}$ (FTS) is effectively a strong TI with a nontrivial weak \mathbb{Z}_2 index, instead of being a perfect weak TI. More precisely, the topological bands in the FTS coexist with other trivial bulk bands that contribute to the bulk Fermi surfaces. This subtle difference causes a breakdown of the QSH picture, which manifests in the “edge states” of the 2D cutting plane. In particular, even though 1D helical modes exist along the dislocation lines, when they come to the top surface, they will merge with a gapless 2D Dirac surface state. This mixed dimensionalities between 1D and 2D Dirac states here make it challenging to directly apply the QSH edge theory in the 2008 PRL paper that the reviewer mentioned. For example, there will be some tricky mathematical issues if one tries to solve for the Majorana wavefunctions, due to the mismatch of dimensionality. Note that if the system is a weak TI with a trivial strong \mathbb{Z}_2 index, then this tricky issue will not show up as the top surface is gapped.

This is essentially why we started to think about an alternative approach to understand the mechanism of dMBS for FTS that can take into account both 1D dislocation helical mode and 2D Dirac surface state. This leads to the development of our nested domain wall picture. While the nested domain wall appears more complicated than the QSH picture, we believe that it is a more accurate description of the dMBSs. For example, we can exploit the nested domain wall picture to analytically extract the wavefunctions and the localization lengths of the dMBSs, which are consistent with the numerical findings. These new analytical results are now included in both the main text and the supplemental material. In the revised manuscript, we have also added a short discussion to clarify the above two pictures, and further included the interesting references that the reviewer has pointed out.

For your information, we have summarized our analytical theory below, which hopefully will provide a better illustration of the nested domain wall strategy.

- ✓ For the topological surface state, we obtain the localized mode near $z = 0$ boundary between sample ($z \geq 0$) and vacuum ($z < 0$)

$$\psi^\uparrow(z) = 2ce^{-Az} \sin(Bz) |\phi_-\rangle \otimes |\uparrow\rangle, \quad (\text{R12a})$$

$$\psi^\downarrow(z) = 2ce^{-Az} \sin(Bz) |\phi_-\rangle \otimes |\downarrow\rangle. \quad (\text{R12b})$$

where $|c| = \sqrt{\frac{A(A^2+B^2)}{2A^2+B^2}}$ is the normalization factor with $A = \frac{v}{2m_2}$ and $B = \sqrt{2m_{0z}m_2 - v^2}$. The spinor part provides the basis for Dirac surface states, $\{|\phi_-\rangle \otimes |\uparrow\rangle, |\phi_-\rangle \otimes |\downarrow\rangle\}$, where $\sigma_x|\phi_\pm\rangle = \pm|\phi_\pm\rangle$ leads to $|\phi_+\rangle = \frac{1}{\sqrt{2}}(1, 1)^T$ and $|\phi_-\rangle = \frac{1}{\sqrt{2}}(1, -1)^T$. Thus, the surface state Hamiltonian up to linear k order,

$$\mathcal{H}_{surf} = v(k_x s_y - k_y s_x). \quad (\text{R13})$$

Please note that \mathbf{s} are Pauli matrices for the spin degree of freedom. The localization length for the surface state is given by $\xi_{surf} = 1/A = 2m_2/v$, which indicate that smaller spin orbit coupling (v) corresponds to larger localization length.

- ✓ On the sample surface, the ferromagnetism coexists with bulk superconductivity, thus the BdG Hamiltonian for top surface of the sample can be described as

$$\mathcal{H}_{BdG} = v(k_y s_y \gamma_z - k_x s_x \gamma_0) - \mu s_0 \gamma_z + M_z s_z \gamma_z + \Delta_0 s_y \gamma_y, \quad (\text{R14})$$

where γ are Pauli matrices for the particle-hole degree of freedom. Without loss of generality, we consider $M_z > 0$ and Δ_0 . The gap closing happens at $\mu^2 + \Delta_0^2 = M_z^2$, and topological gap is given by $\Delta_{topo} = M_z - \sqrt{\mu^2 + \Delta_0}$. For the topological phase of sample surface, the chiral MZM solution for $\mathcal{H}_{BdG}(k_x = 0, -i\partial_y)$ is

$$\psi(y) = a e^{(-M_z + \Delta_0)y/v} |\phi_+\rangle, \quad (\text{R15})$$

where a is the normalization factor and the spinor part is $|\phi_+\rangle = \frac{1}{2}(1, -i, -i, 1)^T$. It is eigen-state of chiral symmetry $s_y \gamma_z |\phi_+\rangle = -|\phi_+\rangle$, and eigen-state of particle-hole symmetry $s_0 \gamma_x K |\phi_+\rangle = i |\phi_+\rangle$. It is also eigen-state of the k_x term, which gives rise to the dispersion of 1D chiral MZM along the 1D boundary of the sample surface, namely, $\langle \phi_+ | k_x s_y \gamma_z | \phi_+ \rangle = k_x$.

- ✓ We next solve the dislocation MZM by inserting a pair of dislocation lines to the system. As we discussed in the main text, the dislocation line is aligned to the z-direction. There are two main steps.
 - Cut the sample into two parts by the cutting plane expanded by this pair of dislocation lines, as illustrated in Fig. R2 (c) [see the dashed rectangle]. The 1D chiral MZM is also divided into two parts, and near the touching edges, those two 1D chiral MZMs propagate along different direction due to the C_{2z} symmetry [see Fig. R2 (d)].
 - Glue these two 1D chiral MZMs by restoring the lattice, as illustrated in Fig. R2 (e).

After these two steps, we can build an effective two-by-two Hamiltonian consisting of two oppositely propagating 1D chiral MZMs, which are

$$\psi_R(y) \approx e^{(-M_z + \Delta_0)y/v} (1, -i, -i, 1)^T, \quad (\text{R16a})$$

$$\psi_L(y) \approx e^{(M_z - \Delta_0)y/v} (1, i, i, 1)^T. \quad (\text{R16b})$$

This gives rise to

$$\mathcal{H}_{dis} = v k_x \tau_z + \text{Im}[t_c] \tau_y + \text{Re}[t_c] \tau_x, \quad (\text{R17})$$

where the inter-edge coupling t_c is due to the direct hopping ($v k_y s_x \gamma_0 \rightarrow -i v \partial_y s_x \gamma_0$),

$$\begin{aligned} t_c &\approx \int_{-\Delta_y}^{\Delta_y} dy \langle \phi_R | s_x \gamma_0 | \phi_L \rangle \times (e^{(-M_z + \Delta_0)y/v} [-i v \partial_y] e^{(M_z - \Delta_0)y/v}) \\ &\propto -i v \frac{M_z - \Delta_0}{v} = -i(M_z - \Delta_0). \end{aligned} \quad (\text{R18})$$

Please also note that t_c is just a constant if there is no dislocation pairs, however, its sign depends on the position for the dislocation case [c.f., Ying Ran et al., Nature Physics 5,

Figure R6: **Schematic for the analytical solution of surface states, chiral Majorana zero modes (MZMs), and dislocation MZMs.** (a) illustrates the localized Dirac surface states of a topological insulator. (b) showcases the interplay between surface magnetism and bulk superconductivity, resulting in chiral Majorana zero modes (MZMs) represented by blue lines along the hinges of the top and bottom surfaces. (c) displays the cutting plane formed by a pair of dislocation lines. The corresponding top view is also presented on the top surface in (d), where each side features one chiral MZM mode (the triangle indicates the propagating direction). The final step involves gluing these two sections together to restore the lattice, leading to the hybridization between the two chiral MZMs and the emergence of 0D dislocation MZMs.

298–303 (2009)]. For example, these two dislocation lines are separated in real space and locate at $(N_{cx}, N_{cy}, z = 1) \rightarrow (N_{cx}, N_{cy}, z = N_z)$ [line 1] and $(N_{cx} + \Delta_x, N_{cy}, z = 1) \rightarrow (N_{cx} + \Delta_x, N_{cy}, z = N_z)$ [line 2]. The in-plane distance between these two dislocation lines are given by Δ_x . Then, we have

$$t_c = \begin{cases} x \leq N_{cx} \text{ or } x \geq N_{cx} + \Delta_x, & \text{direct in-plane hopping: } -i(M_z - \Delta_0), \\ N_{cx} \geq x \geq N_{cx} + \Delta_x, & \text{inter-layer hopping: } -i(M_z - \Delta_0) \times (-1). \end{cases} \quad (\text{R19})$$

The hopping for spin-orbit coupling acquires a π phase shift on circling the dislocation, which explains the minus above. Therefore, t_c is the mass term for the two 1D chiral MZMs, and changes sign. It leads to the 1D domain wall along the line connecting these two dislocation lines. A 0D MZM, called dislocation MZM in the main text, can naturally appear.

Reviewer #3

Remarks to the Author

In the manuscript, Hu and Zhang studied the trapped Majorana quasiparticles at lattice dislocations of topological iron-based superconductors. The key idea is that when a three-dimensional normal state has a weak band topology together with the bulk superconductivity and surface ferromagnetism, a two-dimensional subsystem spanned by screw or edge dislocations can be considered as an embedded second order topological superconductor hosting Majorana corner modes. The theory is further extended to the system without surface magnetism. The authors also carefully discussed the candidate materials and experimental methods for detecting the Majorana states using STM experiment.

Overall, the manuscript is well-written, and especially, the mechanism generating Majorana modes at lattice dislocation is clearly presented. The implication or relevance of the theoretical idea to real iron-based materials is carefully described by using the model Hamiltonian related to iron-based superconductor.

However, the relation between the weak band topology and dislocation Majorana mode is already known. For example, in [T.L Hughes et al., “Majorana zero modes in dislocations of Sr₂RuO₄”, PRB 90, 235123(2014)], [S. B. Chung et al., “Dislocation Majorana zero modes in perovskite oxide 2DEG”, Scientific Reports 6, 25184 (2016)], dislocation Majorana zero modes are carefully studied and its relevance to real materials is also thoroughly discussed. Although the present manuscript contains new development to a certain level compared to these preceding works, the progress looks like incremental, not at the level of fundamental development. Also, the required conditions to observe the dislocation Majorana modes proposed in this manuscript are quite complicated. Simultaneous presence of weak bulk topology, controlled dislocation, uniform surface ferromagnetism is difficult to be achieved in real materials in a manner that the proposed theoretical idea can actually be observed even in iron-based materials. In this respect, I do not think that the novelty of this paper is at the level of Nature Communications. Let me give a couple of additional questions/comments below.

Repy: We thank the reviewer for his/her carefully reading our work. In particular, we appreciate the reviewer’s positive comments on the clarity of our presentation. However, we respectfully disagree with the reviewer on the assessment of the novelty level. In particular, as we will clarify below, our mechanism for 0D dislocation Majorana bound states (dMBSs) is fundamentally different from those presented in the earlier works. Furthermore, as we have elaborated in Section V of the revised manuscript, all the necessary ingredients of our theory have been experimentally observed in FeTe_{0.55}Se_{0.45} (FTS). In particular, motivated by the reviewer’s comment, we numerically find that the dMBSs are quite robust against spatial magnetization fluctuations. Therefore, we strongly believe that the revised manuscript has indeed met the high standards of Nature Communications.

First of all, the key distinction between our work and the aforementioned PRB and Scientific Reports (SR) papers is the topologies of the BdG ground states. In both PRB + SR works, the superconducting state has a nontrivial weak topology that leads to the dMBSs. In our theory, however, **the bulk superconducting state is topologically trivial due to the s-wave pairing symmetry and the weak topological index is only for the normal-state electrons.** Therefore, our work provides the **first example** of 0D dislocation Majorana modes in a **3D bulk topologically trivial** superconductor, which thus

distinguishes our work from previous discussions. We also realized that we have not emphasized this point enough in the previous manuscript, which potentially leads to confusion for the reviewer to recognize the novelty of our work. Therefore, we have improved the relevant discussions accordingly, along with a new table suggested by Reviewer #1, in the latest manuscript.

Secondly, we respectfully disagree with the reviewer on the subjective comment that “Simultaneous presence of weak bulk topology, controlled dislocation, uniform surface ferromagnetism is difficult to be achieved in real materials in a manner that the proposed theoretical idea can actually be observed even in iron-based materials.”. We have discussed all these conditions for $\text{FeTe}_{0.55}\text{Se}_{0.45}$ in Sec. V.A, where **weak normal-state bulk topology, controlled dislocation, surface ferromagnetism are all experimentally observed**. We note that the *uniformity* of surface ferromagnetism may not be guaranteed naturally in FTS. In practice, we can use the field cooling method to align the magnetizations of possible domains. Interestingly, in the new Supplementary Note 3, we explicitly prove that the dMBSs are still robust even when the magnetism is not spatially uniform.

Figure R7: **Robustness of dislocation Majorana zero modes (MZMs)**. We consider a spatial magnetization fluctuation $\widetilde{M} \in [-\Delta_{M_z}, \Delta_{M_z}]$ in the full 3D lattice calculation for the energy of dislocation MZM and bulk states. Specifically, we focus on the z -component magnetization and find dislocation MZMs are stable even before Δ_{M_z} exceeds 5 times the value of M_z . This critical value implies the break down of proximity-induced surface superconductivity.

In particular, we consider a spatial fluctuation $\widetilde{M} \in [-\Delta_{M_z}, \Delta_{M_z}]$ so that the total magnetization $M_z + \widetilde{M}$ is considered in our system. For a full 3D lattice calculation, the value of \widetilde{M} is random for each lattice site. For every configuration, we calculate the lowest four energies: $\{E_{MZM}, E_1, E_2, E_3\}$ using the sparse matrix method. The average of magnetization

fluctuations is then performed as

$$E_{MZM}^{avg} = \frac{1}{N_{avg}} \sum_{\text{configurations}} E_{MZM}, \quad (\text{R20a})$$

$$E_1^{avg} = \frac{1}{N_{avg}} \sum_{\text{configurations}} E_1, \quad (\text{R20b})$$

$$E_2^{avg} = \frac{1}{N_{avg}} \sum_{\text{configurations}} E_2, \quad (\text{R20c})$$

$$E_3^{avg} = \frac{1}{N_{avg}} \sum_{\text{configurations}} E_3. \quad (\text{R20d})$$

In the simulation, we use $N_y = 2N_x = 2N_z = 32$ and consider only top surface magnetism for simplicity. Here N_{avg} is the number of fluctuating magnetic configurations and $N_{avg} = 20$ is found to be large enough for our purpose. The numerical results for MZM (E_{MZM}^{avg}) and lowest bulk energies (E_2^{avg} , E_3^{avg} and E_4^{avg}) are shown in Fig. R7. Remarkably, when $\Delta_{M_z} < 3M_z$ with $M_z = 1$, the system is highly robust against the magnetism fluctuations, as shown in Fig. R7. In fact, **the dislocation Majorana modes are spoiled only when Δ_{M_z} exceeds 5 times the value of M_z , which we believe is unlikely to happen in experiments.** This is because when M_z is extremely large, the dislocation MZMs are not stable due to the pair-breaking effect of proximity-induced surface superconductivity. This limit, however, is inconsistent with the experimental observations for FTS by angle-resolved photoemission spectroscopy [c.f. Proceedings of the National Academy of Sciences 118 (3), e2007241118 (2021) and Nat. Mater. 20, 1221–1227 (2021)], where surface magnetization indeed coexists with superconductivity. This new result clearly proves the robustness of dMBSs against the magnetism fluctuations in real space. This is why we are confident that our proposal will be realized in experiments very soon.

A Point-to-Point Response to Additional Comments

Comment 1: In Figure 1 (b), it is better to mark the axes for real space coordinates. In Figure 1 (c), is $t=t(x)$ correct? $t=t(y)$ looks suitable.

Reply: We thank the reviewer for this great suggestion. We have added the axes in Fig. 1 and replaced $t = t(x)$ to $t = t(y)$.

Comment 2: On page 3, in the paragraph starting with “To explore the fate of chiral Majorana hinge modes during the gluing process, . . .”. The third sentence should be “Distribution of δ_M and δ_{SC} are shown in Fig.1(c).”

Reply: We thank the reviewer for pointing out this typo. We have fixed it in the revised manuscript.

Comment 3: Related to the sentence “However, the interlayer mass term for the Majorana fermions will obtain a phase factor $e^{i\pi b^* n u}$, following the side Dirac surface states”, I suggest to add a couple of sentences describing the origin of the phase factor.

Reply: We thank the reviewer for this suggestion. We have revised our manuscript accordingly to improve this part.

Comment 4: The meaning of the term “dislocation dipole” should be more clearly presented.

Reply: We thank the reviewer for pointing out this issue. “dislocation dipole” is a known terminology in the crystal defect community and we have added a short definition to clarify possible confusions to the general readers.

Comment 5: The description of the inflation of a Majorana mode from 0D to 1D seems to be valid only when the circular radius of the cylindrical geometry is small. I do not understand how meaningful this description is in bulk materials.

Reply: This is an interesting question! We are motivated to study this problem by a purely conceptual question: what will happen if the cutting plane of a dislocation terminates at the system’s boundary, but not at another dislocation? Given that Majorana modes must come in pairs (i.e., the fermion doubling theorem) and the fact that a Majorana mode can only annihilate with another Majorana mode, the only logical outcome would be the “inflation” picture. We agree with the reviewer that this practical effect of this phenomenon in the thermodynamic limit could be limited, as the chiral Majorana hinge mode would have a vanishingly small gap when $R \rightarrow \infty$. Nonetheless, this effect is of conceptual interest since it directly suggests that the branch cut of the dislocation mimics the effect of a quantized magnetic flux (e.g., a vortex).

Comment 6: Related to the Figure 3 and related description, it is stated that PFS and ET₂ phases can coexist. However, when the bulk is gapless, I am wondering how stable the Majorana modes are and how their spatial localization is affected by. Also, what is the qualitative difference between Figure 3 (c) and (d)? It is unclear what kind physics is intended to be delivered by comparing these two panels.

Reply: We thank the reviewer for this question. Indeed, when PFS and ET₂ coexist, the dislocation Majorana mode could interact with the gapless background, making it much easier to hybridize with another Majorana mode at a different dislocation. So we agree with the reviewer that this is an “unstable” situation for Majorana modes. Fortunately, most experiments for the bulk FTS samples that we are aware of are supporting a dominating out-of-plane magnetization, which promotes an ET₂-only phase in our phase diagram.

Interestingly, a recent experimental work by Gang Qiu et al. [Nat. Commun. 14, 6691 (2023)] on the thin film samples also shows evidence of both in-plane magnetization and PFS.

Given that PFS is now experimentally accessible and we thus find it necessary to clarify the two distinct scenarios: (i) PFS with a trivial Chern number in Fig. 3 (c); and (ii) PFS with a nontrivial Chern number in Fig. 3 (d). Therefore, we include both results (i.e., Fig. 3 (c) and (d)) in the main text, just for completeness.

Comment 7: The discussion about class DIII ET₂ is rather brief. How the bulk $s_{+,-}$ pairing can induce a pairing mass domain for Dirac fermions?

Reply: The mechanism for inducing a surface-dependent pairing domain wall with s_{\pm} pairing is introduced in Ref. [64]. The domain wall occurs as a consequence of the interplay between the band inversion at Z and the unconventional nature of the s_{\pm} pairing. One could easily verify this picture by analytically solving the Dirac surface states at different surfaces and projecting the bulk pairing onto the surface-state basis.

Starting from the helical hinge Majorana modes, it is quite straightforward to perform a similar nested-domain-wall argument to understand the class-DIII ET₂ phase. We think that Fig. 4 (a) and (b) should be self-explanatory for readers who have followed our detailed discussions in Sec. II. Therefore, we believe Sec. IV is already complete as it is.

Comment 8: As for the CdGM states inside the SC gap, it is stated that ET₂ dislocation core carries fewer or even no subgap states besides the dMBS. What is the physical reason for it?

Reply: We thank the reviewer for this interesting question. This arises from the different microscopic mechanisms for vortex bound states and dislocation bound states, which we summarize below:

- *Vortex Bound States:* The solution of CdGM shows that the energy of vortex bound states in a s-wave superconductor is about $l_z \frac{\Delta_0}{E_f^2}$ with l_z the angular momentum. The superconducting gap Δ_0 is typically smaller than the Fermi energy E_f , which leads to a series of in-gap bound states.
- *Dislocation Bound States:* The dislocation Majorana modes, however, is an outcome of multiple Jackiw-Rebbi processes. Given that the conventional Jackiw-Rebbi problem does not necessarily lead to other finite-energy bound states beyond the zero mode, it is then not surprising that the dislocation core will feature fewer or no subgap states.

We have revised our manuscript to further clarify this difference.

Comment 9: I think that more discussion is necessary related to the spatial anisotropy of dMBS wave function. How is the anisotropic shape related to the in-plane mirror symmetry? I think the same symmetry breaking effect also exists in vortex Majorana wave function that is usually circular symmetric. I think the issue is the length scale of the spatial localization of the wave function, not related to whether the Majorana is bound to a dislocation or a magnetic vortex.

Reply: We thank the reviewer for this inspiring comment! We agree with the reviewer that in practice, the shape of vortex MBS may not be perfectly circular as it can be affected by many factors, including the underlying lattice group, orientation of the magnetic field, and the surface impurities. We have revised the manuscript accordingly to clarify possible confusions.

Comment 10: How the non-Abelian nature of the dislocation Majorana zero mode can be experimentally detected?

Reply: This is an interesting question. One will need a probe with an atomic-scale resolution to capture any potential signal of dMBS, and the best probe we currently have is the scanning tunneling microscopy (STM). Unfortunately, STM is not good at reading phase-sensitive signals. Thus how to verify non-Abelianity using STM has remained a highly challenging open question in the Majorana community. Notably, the exactly same question also remains open for the vortex Majorana systems. At this point, we believe that observation of robust zero-bias peak will provide a strong evidence for dMBS, which unfortunately cannot be treated as an unambiguous smoking gun. Proposing a practical measurement protocol of the non-Abelian nature of dMBS is well beyond the scope of this work, but will be certainly of future research interest.

Comment 11: It is stated that ET₂ phase is “extrinsic”. Why is it impossible to characterize dMBS by 2D bulk topological invariant of the subsystem? Any fundamental reason for it?

Reply: This is a great question! Indeed, we can define a 2D inversion-symmetry Z_4 indicator κ for the 2D cutting plane, which follows the definition in Phys. Rev. Research 2, 013064 (2020). In the ET₂ case, the “bulk” topological index κ will be found to be trivial. From a different perspective, an inversion-protected 2nd-order bulk topological superconductor will have two Majorana corner modes in total. For the ET₂ phase, however, there exist four dMBSs, each sitting at the corner of the cutting plane. Therefore, the ET₂ phase is not bulk topological, while it belongs to the “extrinsic” 2nd-order topological phase that is protected by the edge gap. In some literature, this phase is also known as a “boundary-obstructed” topological state. Recently, we have studied an example of the intrinsic ET₂ phase, but its microscopic mechanism is fundamentally different from the extrinsic one presented in this work. We leave this discussion to future work.

A list of changes in blue

□ Main text.

- ✓ On page 1, we revise the introduction to clarify our motivation of this work.
- ✓ On page 2, below Sec II, we cite those papers suggested by reviewers for the nested domain wall construction.
- ✓ On page 2, in Sec. II (A), we emphasize the Dirac surface state is two fold degenerated.
- ✓ On page 3, in Fig. 1, we change “weak topology” to “weak \mathbb{Z}_2 index” (also emphasized in the figure caption), and refine the typo in (c).
- ✓ On page 4, above Sec. II (B), we add one paragraph to discuss the localization length of dislocation Majorana zero modes.
- ✓ On page 4, in Sec. II (B), we define the dislocation dipole to avoid confusions to readers, and add two sentences to discuss the gauge-dependence of dislocation planes.
- ✓ On page 5, we add a table to clarify the topological physics for bulk normal states, bulk superconducting states, and various boundary modes.
- ✓ On page 5, above Sec. III, we add one paragraph to clarify that the ET_2 phase in our work is extrinsic, compared to previous works.
- ✓ On page 6, in the end of Sec. III (B), we add one sentence to comment on the Majorana mode and partial Fermi surface states.
- ✓ On page 8, in Sec. V (A), we add new experiment paper to support our theory.
- ✓ On page 8, in the end of Sec. V (A), we add one paragraph to emphasize that the new results from the DFT-based model is in good agreement with results in Fig. 3.
- ✓ On page 9, in Sec. V (C), we comment on the realization of 0D dislocation Majorana mode in weak TI, and clarify the importance of nested domain wall constructions for strong TI with weak \mathbb{Z}_2 index.
- ✓ On page 10, in Sec. VI, we add one sentence to discuss there is no or fewer in-gap bound states in our system.
- ✓ On page 10, in Sec. VI, we add one paragraph to clarify the wavefunction distribution of dislocation Majorana mode, and compare it to vortex Majorana mode.
- ✓ On page 10, in Sec. VI, we discuss the robustness of dislocation Majorana mode against spatial magnetization fluctuations.
- ✓ On page 10, in Sec. VII, we emphasize the novelty of this work, namely, the bulk superconducting state is trivial in our study for Majorana mode.
- ✓ On page 11, in Sec. VII, we remove the previous discussion of “defect-obstruction”, but add one paragraph to discuss potential future directions based on our work.

□ Supplementary Note.

We add a 14-page long supplementary note to support the discussions and major revisions made in the revised manuscript. (1) We derive low-energy effective theory; (2) We

use analytical and numerical methods to explore the localization length of dislocation Majorana modes.; (3) We add the discussion of DFT-based model for FeTeSe, including the model Hamiltonian, superconducting bands, the topological phase diagram that supports the ET_2 phase discussed in the main text; (4) We discuss the robustness of dislocation Majorana against surface magnetization fluctuations.

Below is the title of the supplementary note:

- ✓ Supplementary Note 1. Localization Length of Dislocation Majorana Bound States
 - 1.1 Analytical theory for nested domain wall construction
 - 1.2 Numerical approach I: finite-size gap
 - 1.3 Numerical approach II: wave functions
- ✓ Supplementary Note 2. Surface Topological Phase Diagram for $FeTe_{1-x}Se_x$: a First-Principles-Based Model Study
 - 2.1 Eight-band model Hamiltonian and parameters
 - 2.2 Superconducting band structures
 - 2.3 Surface topological phase diagrams
 - 2.4 In-plane magnetism induced partial Fermi surfaces
- ✓ Supplementary Note 3. Robustness of dMBS Against Magnetic Disorders

REVIEWERS' COMMENTS

Reviewer #1 (Remarks to the Author):

I appreciate the strong effort that the authors have made to respond to all of my previous comments. I find the revised manuscript to be substantially improved and to largely be suitable for publication in Nature Communications. I have one minor comment, which is detailed below. After this comment is implemented, I recommend this work for publication without further review from myself.

Minor comment:

In their response to my previous recommendation that the authors use DFT calculations to realistically model the dislocation Majorana bound states (MBS), the authors note in the supplementary material and response letter that using realistic tight-binding parameters, the z-direction MBS localization length would likely be hundreds of unit cells, precluding their simulation using the authors' computational resources. The authors also note that using realistic parameters, the Dirac surface state of the normal-state system has a localization length of around 30 layers, which can be contrasted with the much shorter surface-state localization lengths in bismuth selenide and telluride. Both of these points are very important, but neither of them, as far as I can discern, appears in the current iteration of the main text.

In their final revision of the manuscript, I recommend that the authors add a few sentences to the main text emphasizing these points and discussing their experimental implications. For example, this would seem to suggest that few-layer samples, and potentially even thin films, would be poor venues for realizing the MBS predicted in this work, due to the long localization lengths of the topological bound states.

Reviewer #2 (Remarks to the Author):

I want to thank the authors for clarifying my point regarding the generation of Majorana bound states from the quantum spin-Hall edge modes formed at a dislocation loop. Indeed, I had overlooked the simultaneous presence of weak and also strong topology, which renders the full surface gapless. My recommendation remains unchanged, I believe that this work should be published in Nature Communications.

Reviewer #3 (Remarks to the Author):

I think the authors for their great effort to answer my question. In the previous round of the review, my main concern was about the novelty of the present work compared to previously published papers as well as the relevance of the proposed theory to real experiments. I think the authors properly answered these questions with suitably revision of the manuscript. I recommend the publication of the revised manuscript in Nature Communications.

Reviewer #1 (Remarks to the Author):

I appreciate the strong effort that the authors have made to respond to all of my previous comments. I find the revised manuscript to be substantially improved and to largely be suitable for publication in Nature Communications. I have one minor comment, which is detailed below. After this comment is implemented, I recommend this work for publication without further review from myself.

Reply: We thank the Reviewer #1 for his/her recommendation.

Minor comment:

In their response to my previous recommendation that the authors use DFT calculations to realistically model the dislocation Majorana bound states (MBS), the authors note in the supplementary material and response letter that using realistic tight-binding parameters, the z-direction MBS localization length would likely be hundreds of unit cells, precluding their simulation using the authors' computational resources. The authors also note that using realistic parameters, the Dirac surface state of the normal-state system has a localization length of around 30 layers, which can be contrasted with the much shorter surface-state localization lengths in bismuth selenide and telluride. Both of these points are very important, but neither of them, as far as I can discern, appears in the current iteration of the main text.

In their final revision of the manuscript, I recommend that the authors add a few sentences to the main text emphasizing these points and discussing their experimental implications. For example, this would seem to suggest that few-layer samples, and potentially even thin films, would be poor venues for realizing the MBS predicted in this work, due to the long localization lengths of the topological bound states.

Reply: We thank the Reviewer #1 for this suggestion. We added this discussion in the Methods section.

Below Eq. (12), we add "In Supplementary Note 2. A, we numerically find that the localization length of the Dirac surface state in $\text{FeTe}_{0.5}\text{Se}_{0.5}$ is about 30 layers (~18 nm). This localization length is much larger than the typical scale observed in Bi_2Se_3 and Bi_2Te_3 (~3 nm), as the spin-orbit coupling strength in $\text{FeTe}_{0.5}\text{Se}_{0.5}$ is smaller. Therefore, the $\text{FeTe}_{0.5}\text{Se}_{0.5}$ sample should be at

least 60 layers in thickness along (001) direction, in order to observe the predicted dislocation MBS signals.”

Reviewer #2 (Remarks to the Author):

I want to thank the authors for clarifying my point regarding the generation of Majorana bound states from the quantum spin-Hall edge modes formed at a dislocation loop. Indeed, I had overlooked the simultaneous presence of weak and also strong topology, which renders the full surface gapless. My recommendation remains unchanged, I believe that this work should be published in Nature Communications.

Reply: We thank the Reviewer #2 for his/her recommendation.

Reviewer #3 (Remarks to the Author):

I think the authors for their great effort to answer my question. In the previous round of the review, my main concern was about the novelty of the present work compared to previously published papers as well as the relevance of the proposed theory to real experiments. I think the authors properly answered these questions with suitably revision of the manuscript. I recommend the publication of the revised manuscript in Nature Communications.

Reply: We thank the Reviewer #3 for his/her recommendation.